



# Investigating the dynamics of floating wind turbine wakes under laminar inflow using large eddy simulations

Ricardo Amaral[1], Felix Houtin-Mongrolle[2], Dominic von Terzi[1], Kasper Laugesen[3], Paul Deglaire[4], and Axelle Viré[1]

[1]Delft University of Technology, Wind Energy Section, Kluyverweg 1, 2629 HS Delft, Netherlands
[2]Siemens Gamesa Renewable Energy (SGRE), Prinses Beatrixlaan 800, 2595 BN Den Haag, Netherlands
[3]Siemens Gamesa Renewable Energy (SGRE), Borupvej 16, 7330 Brande, Denmark
[4]Siemens Gamesa Renewable Energy (SGRE), Avenue de l'Université C/O Insa Rouen - BP 08, 76800 Saint Étienne de Rouvray, France

**Correspondence:** Ricardo Amaral (r.pintoelisbaomartinsamaral@tudelft.nl)

**Abstract.** A laboratory-scale model of the DTU 10 MW wind turbine is investigated under prescribed surge, sway, roll, pitch, yaw and coupled surge-pitch motions using YALES2, a high-fidelity large-eddy simulation (LES) tool coupled to an actuator-line model. The prescribed motions are sinusoidal and two cases per degree-of-freedom (DOF) are considered: one with a low Strouhal number $St$ and high normalized amplitude $A^*$, and vice versa. The turbine is operated in the vicinity of rated conditions, under constant rotor speed and steady uniform free-stream velocity. The different aspects of the methodology are validated including experimental comparison with data from the OC6 Phase III campaign. Cases with low-$St$/high-$A^*$ exhibit behavior similar to the fixed-bottom case in many aspects of the wake. Conversely, cases with a high-$St$/low-$A^*$ disturb the wake to a much larger extent. The contrast is caused by differences in how much the wake amplifies the perturbations of the floating motion upstream and is particularly noticeable in the blade tip and root shear layers (a.k.a. turbulent mixing layers), where a higher amplification leads to a faster expansion of the wake in the high-$St$/low-$A^*$ cases. Prescribed motions with a component perpendicular to the flow are found to have a larger impact than motions exclusively in the flow direction. The prescribed motions also cause wake meandering, with larger amplitudes observed for the cases with low-$St$/high-$A^*$. The magnitude of perturbation amplification is quantified by amplification factors relating either the wake velocity or displacement to the rotor center velocity or displacement, respectively. The maximum factors are associated with the high-$St$/low-$A^*$ cases and are similar among DOFs, with some exceptions.

## 1 Introduction

In September 2024, Europe had 35 GW of installed offshore wind power capacity, which is set to grow to 54 GW by 2030 (Costanzo et al., 2024). Most of the offshore wind turbines installed so far are fixed-bottom wind turbines (FBWTs), since they are more cost-competitive than floating offshore wind turbines (FOWTs) for water depths below 50-60 m (Cozzi et al., 2019; Barooni et al., 2023). However, harvesting wind energy in locations with water depths higher than 60 m, and more than 60 km away from the shore, has the potential to supply the world's total electricity demand several times by 2040 (Cozzi et al., 2019).



At these water depths, FOWTs are more cost-competitive, making the technology a serious candidate to support the energy transition. As a result, FOWT farms are already under operation, deployment and auction, with 10 GW of capacity expected in Europe by 2030 (WindEurope, 2022). More than ever, it is necessary to have an holistic and in-depth understanding of
FOWTs, their physics and the way they interact with each other. These can differ materially from FBWTs, since the rotor is free to move. Engineering models widely used for wind turbine load design are based on blade-element momentum theory (BEMT) and rely on assumptions that, while reasonable for FBWTs, may break down for FOWTs (see, for example, Sebastian and Lackner (2013); Farrugia et al. (2016); Kyle et al. (2020); Micallef and Rezaeiha (2021); Ramos-García et al. (2021); Papi et al. (2024)). On a similar note, engineering wake models that do not intrinsically solve the time-varying nature of the wake
(Jensen, 1983; Katic et al., 1986; Ainslie, 1988; Larsen, 2009; Bastankhah and Porté-Agel, 2014), may miss wake perturbations caused by the platform motion in a FOWT. Even the dynamic wake meandering model (Larsen et al., 2007) that captures this time dependence assumes that a base stationary wake is convected as a passive scalar subjected to large-scale ambient turbulent structures, thus capturing only one of the wake meandering processes. Due to the rotor motion in FOWTs, the wake is subject to perturbations that can be amplified by the wake and alter its behavior, as demonstrated experimentally and computationally
by Mao and Sørensen (2018), Li et al. (2022) and Messmer et al. (2024), for instance. This behavior change is most evident at the tip and root shear layers and in the wake meandering, with the process being highly nonlinear, unsteady, and dependent on the perturbation characteristics. The term "shear layer" (a.k.a. "turbulent mixing layer") is used throughout to refer to the vicinity of the wake boundary that separates the wake interior from the free flow and that is affected by vortices shed at the blade extremities. To the authors' best knowledge, state-of-the-art low-fidelity wake models cannot capture these phenomena
and further investigation is necessary to assess if these phenomena are important and, if so, how to incorporate them into the low-fidelity wake models. Due to the relevance of this topic for wind farm operation, it has been investigated over the past two decades, both experimentally and numerically.

## 1.1 Wake shear layers and meandering for fixed-bottom wind turbines

The first investigations were naturally performed on FBWTs. Medici and Alfredsson (2006) tested a wind turbine with hot-wire
anemometry in the wind tunnel and identified low-frequency content in the wake spectra associated with wake meandering and consistent with bluff body vortex shedding. Kang et al. (2014) performed large-eddy simulations (LES) on a hydrokinetic turbine and identified tip and hub counter-rotating shear layers that merged at approximately $x^* = 3$ (where $x^* = x/D$ and $D$ is the rotor diameter), triggering a steep and strong increase in turbulence intensity ($TI$) at the outer wake layer. This region also corresponded to the onset of wake meandering. The tip shear layer was further studied by Lignarolo et al. (2015) who
conducted stereoscopic particle image velocimetry to map the velocity fields of the tip-vortices of a small wind turbine. The kinetic energy balances revealed that the energy entrainment was negligible and turbulence production was low in the near-wake, while both increased sharply after the tip-vortex leapfrogging phenomenon, confirming the prediction by Medici (2005). The wake frequency content was investigated by Heisel et al. (2018) who acquired field measurements on a wind turbine. The wake energy spectra had lower frequency content at Strouhal numbers $St < 0.1$ and higher frequency content at $St > 0.1$
for downstream distances between $x^* = 2.6$ and $x^* = 3.1$, when compared to those of the undisturbed flow. The compilation





and comparison with previous experimental results (Medici and Alfredsson, 2006; Chamorro and Porté-Agel, 2010; Chamorro et al., 2013; Okulov et al., 2014; Howard et al., 2015) showed that the peak $St$ frequency tends to be within $0.1 - 0.4$ and is $Re$-independent, where $Re$ is the Reynolds number. De Cillis et al. (2020) performed LES on a turbine, both with and without nacelle and tower. Without the nacelle and tower, a proper orthogonal decomposition (POD) analysis indicated that the strongest modes were associated with the tip and root vortices and that they almost always transferred energy from the wake into the shear layer, slowing the recovery down. Gambuzza and Ganapathisubramani (2023) showed experimentally for a lab-scale wind turbine under Kolmogorov-like flows, that higher $TI$ values cause a faster transition to the far-wake, faster recovery and a smaller wake length. They observed that the main driver for wake meandering and wake transition is the onset of the shear-layer instability that is exacerbated by the free-stream turbulence.

Previous research identified the shear layer and wake meandering as natural phenomena in wind turbine wakes. These studies also identify links between both as well as with the wake recovery. The research also established that the near-wake tip vortex sheet isolates the wake from the free flow, preventing momentum exchange and slowing down the wake recovery.

### 1.2 Wake shear layers and meandering for floating offshore wind turbines

In a FOWT, the wake will evolve subjected to an upstream perturbation caused by the floating turbine structural motion. This motion can modulate the wake characteristics in different ways, depending on the structural motion characteristics. The wake behavior may thus significantly depart from that observed in FBWT and many authors deepened the literature in this topic. Mao and Sørensen (2018) simulated an actuator disk with an algorithm based on the incompressible Navier-Stokes equations, subjected to a spatially and temporally variable inflow perturbation. The perturbations in the domain were maximized for certain pairs of inflow perturbation amplitude and frequency. Perturbations were mainly in the streamwise and radial directions and were contained within the turbine radius. For those pairs, the dominant frequency, at least up to $x^* = 5$, was the inflow frequency. They concluded that the inflow perturbations were amplified in the tip shear layer, while expanding radially with the downstream wake advection, leading to streamwise velocity oscillation magnitudes at the centerline at $x^* = 5$ that were one order of magnitude above the inflow perturbation. Ramos-García et al. (2021) used an aeroelastic vortex-solver to simulate the IEA Wind 15-MW reference wind turbine in laminar flow under prescribed surge and pitch motions. At below-rated conditions, the pitch frequency parametric study with $A = 6.38$ deg revealed that it promoted wake recovery and turbulence intensity, especially for $St = 0.42$. The pitch amplitude parametric study at $St = 0.73$, suggested that higher amplitudes lead to faster recovery and higher $TI$ values. The surge results were very similar to the pitch results. For above-rated conditions, the pitch frequency parametric study came in contrast with the one performed at below-rated conditions in that the wake recovery increased marginally while the case with $St = 0.27$ severely hindered this process. The $TI$ levels were still far higher than the fixed-bottom case for all pitch cases. The pitch amplitude parametric study at $St = 0.39$, suggested that higher amplitudes slow the recovery at $x < 10D$ and speed it up afterwards. $TI$ values were again higher for all cases. The surge results were once more very similar to the pitch results. Li et al. (2022) used LES to simulate the NREL offshore 5 MW reference wind turbine under prescribed sway motion. Their analysis showed that sway perturbations with frequencies in the $St$ range of $0.2 - 0.6$ and $A/D$ in the range of $0.01 - 0.04$ were amplified by the wake and led to wake meandering that can be of the order of



the rotor diameter. The wake meandering was accompanied by a faster wake recovery and the wake streamwise profile was highly sensitive to the $St$ and $A/D$. $St$ below 0.1 had virtually no effect on the recovery. For $St = 0.1 - 0.3$, larger amplitudes fostered the recovery, while for $St \geq 0.5$, the recovery was hindered. As for the wake turbulence intensity $TI$, all cases led to large increases in $TI$, except for one case at the lowest $St$. All the $TI$ profiles as a function of the downstream position were characterized by an increase up to a peak, followed by a decrease, with the peak position being highly sensitive to both $St$ and $A/D$. For inflow $TI$ values below $1.5\%$, the side-to-side motion drove the wake evolution while for values above $10.6\%$, the inflow turbulence dominated. In between, both effects were coupled. Belvasi et al. (2022) experimentally tested a porous disk on an atmospheric boundary layer with $TI = 8\%$ at hub height. The wake analysis focused on the position at $x^* = 8.125$. The pitch case at $St = 0.28$ and $A = 8$ deg slightly improved the wake recovery relative to the fixed-bottom case, especially for the bottom part of the disk. The remaining pitch and surge cases where performed at $St < 0.14$ and led to a slightly worse recovery. The standard deviations of the wake centers of the floating cases were consistently higher than those of the fixed case. Also, the standard deviations in the side-to-side direction were much higher than those in the height direction. Spectral analysis of the wake power revealed a peak at the prescribed $St$ for the pitch and surge cases. Messmer et al. (2024) demonstrated experimentally that single-DOF prescribed motions in surge and sway enhance the wake recovery in up to $25\%$ relative to the fixed-bottom case, and especially for $St$ in the range of $0.2 - 0.6$ and $0.3 - 0.9$, respectively. For sway, the recovery improvement showed a maximum for $St \approx 0.4$ at several downstream positions while for surge, the recovery was maximized over a range. Prescribed motion amplitude dependency was also identified. Furthermore, they showed that these motions work by shifting the streamwise position at which the recovery gradient increases sharply, which coincided approximately with the point where the shear layers merge and the wake transitions to the far wake. The mechanism identified as the enhanced recovery drivers were meandering structures for sway. For surge, depending on the $St$, the recovery was driven by pulsating structures or a combination of meandering and pulsating structures. Wakes from both DOFs were found to be similar to that of the fixed-bottom turbine for $St < 0.1$. The power spectra values of the prescribed motion frequency tend to increase up to a peak, often located between $x^* = 2$ and $x^* = 6$, and decrease therefrom. At the shear layer and contrary to decaying turbulence, the inertial range of the energy spectra broadens as the wake progresses downstream which points to energy being fed into the flow in this region. The centerline spectra at the recovery onset position were found to collapse into one, indicating wake universality at this point. Last but not least, it is important to refer that the floating motion effects on wakes are relevant for the dynamics and power production of downstream turbines (see for instance Wise and Bachynski (2019); Ramos-García et al. (2022); Duan et al. (2023); Mou and Porté-Agel (2024)).

All these investigations demonstrate that upstream periodic perturbations within certain frequency and amplitude ranges modulate the wake by amplifying its meandering and anticipating the shear layer breakdown. This, in turn, accelerates the wake recovery with an accompanying increase in $TI$. It was also shown that ambient turbulence has a detrimental effect on the perturbation amplification. This factor seems to be the main driver of the recovery for mid-to-high $TI$ values, at which it severely disrupted the floating motion-induced wake structures.





### 1.3 Objective

All in all, the literature suggests that upstream wake perturbations arising from the FOWT structural motion can have a signifi-
cant impact on their wake evolution and on downstream turbines, which is highly dependent on the perturbation characteristics
and ambient turbulence. Wake perturbation amplification appears to be especially important at low $TI$ values, which can occur
offshore under stable atmospheric conditions. Although the phenomenology of the wake perturbation evolution is investigated
in the literature for motions in a small set of DOFs, a comparison between a large set of motions in several DOFs at the same
time is still missing. Hence, this paper investigates a small-scale wind turbine model under prescribed sinusoidal surge, sway,
roll, pitch, yaw, and simultaneous surge and pitch in order to assess the potential effects that a floating wind turbine motion
may have on the wake evolution and the differences between each DOF.

## 2 Methodology and setup

### 2.1 Numerical framework

Studying wind turbine wakes in detail requires the use of high-fidelity techniques at high mesh resolutions to capture the small
scales that may be determinant in driving the wake evolution. LES provide a suitable method that consists on spatially filtering
the Navier-Stokes (NS) below a filter size somewhere in the inertial range. For a constant density fluid, the filtered NS equations
are:

$$\frac{\partial \widetilde{u}_j}{\partial t} + \frac{\partial \widetilde{u}_i \widetilde{u}_j}{\partial x_i} = -\frac{1}{\rho}\frac{\partial \widetilde{p}}{\partial x_j} - \frac{\partial \tau_{ij}^{\mathrm{r}}}{\partial x_i} + \nu \frac{\partial^2 \widetilde{u}_j}{\partial x_i \partial x_i} + \widetilde{f}_j \qquad \text{and} \qquad \frac{\partial \widetilde{u}_i}{\partial x_i} = 0, \tag{1}$$

where Einstein's notation is used, $\widetilde{\cdot}$ is the filtering operator, $u$ is the velocity vector, $p$ is the pressure, $\nu$ is the kinematic
viscosity, $\rho$ is the density and $f$ the external volume forces. The subgrid scale stress tensor $\tau_{ij}^{r}$ is based on the Dynamic
Smagorinsky model (Germano et al., 1990). The actuator-line model (ALM) is used to represent the turbine, where each blade
is discretized in a collection of actuator points with an associated 3D body force that acts on the filtered NS equations as a
velocity source term. This set of equations is integrated using the YALES2 flow solver (Moureau et al., 2011). YALES2 is
a massively parallel finite-volume solver, which is specifically tailored for LES, and relies on a central 4th-order numerical
scheme for spatial discretization, and a method similar to the 4th-order Runge-Kutta method (Kraushaar, 2011) for the time
integration on unstructured grid. YALES2's ALM implementation is described by Houtin-Mongrolle (2022) and the necessity
of fairly high order numerics to ensure the proper transport of fine vortical structures, in the context of ALM, was demonstrated
by Benard et al. (2018).

### 2.2 Wind turbine model and floating degrees-of-freedom

The wind turbine model used is the one utilized in the UNAFLOW project and consists of a laboratory-scale DTU 10 MW wind
turbine (Fontanella et al., 2021a, b; Bak et al., 2013). The model was designed to have the same thrust curve as the real-scale



DTU 10 MW, at lower rated wind speed, by means of low-$Re$ number airfoils (Bayati et al., 2017). Some standard features of the model can be seen in Table 1. Figure 1 (a) shows the model in perspective, where the rotor rotates clockwise.

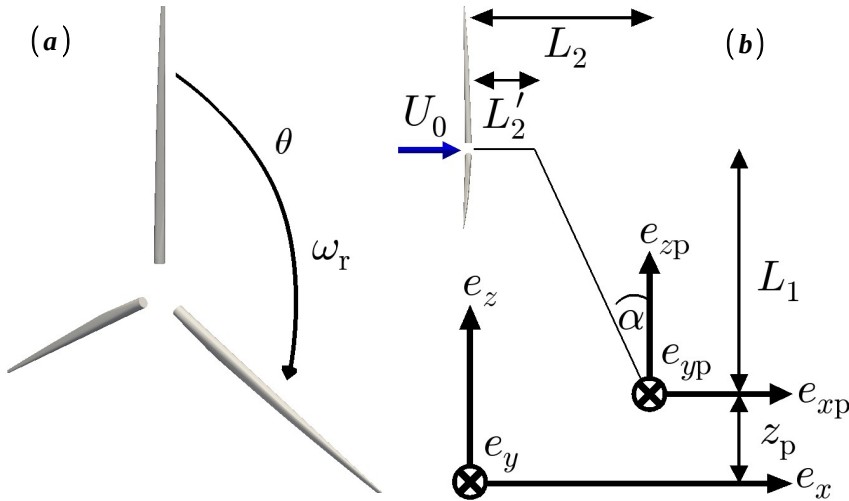

**Figure 1.** Simulated components (blades) in perspective (a) and initial turbine setup with reference frames (b).

**Table 1.** Turbine characteristics and operating conditions.

| Rotor diameter | Hub diameter | Blade length | Rotor tilt | Free-stream velocity | Rotor speed | Tip-speed ratio |
|---|---|---|---|---|---|---|
| $D$ | $D_{\mathrm{h}}$ | $L_{\mathrm{b}}$ | - | $U_0$ | $\omega_{\mathrm{r}}$ | $\lambda$ |
| 2.381 m | 0.178 m | 1.102 m | 5 deg | 4 m s$^{-1}$ | 240 rpm | 7.476 |

**Table 2.** Turbine setup dimensions.

| | $z_{\mathrm{p}}$ | $L_1$ | $L_{2'}$ | $L_2$ | $\alpha$ | $z_{\mathrm{rc}} = z_{\mathrm{p}} + L_1$ |
|---|---|---|---|---|---|---|
| Translation | - | - | - | - | 5 deg | 2.086 m |
| Rotation | 0.730 m | 1.458 m | 0.139 m | 0.267 m | 5 deg | 2.188 m |

The rotor azimuth angle $\theta$ is defined as the azimuth of the blade that is upward pointing at the start of the simulation, i.e., the
155 blade with $\theta_0 = 0\ deg$. The turbine configuration is the same as the one used in the IEA task OC6 Phase III (Bergua et al., 2023; Cioni et al., 2023; Fontanella et al., 2021a, b) and this can be seen in Fig. 1 (b). No tower, nacelle nor hub are simulated but these components are shown in Fig. 1 (b) because their dimensions are taken into account when prescribing motions in rotational DOFs. The setup dimensions are shown in Table 2, where a difference in hub height between translation and rotation cases is present, as per the experimental setup. It is important to notice two things. First, the rotor tilt and the angle $\alpha$ cancel



out so that the rotor is perpendicular to the free-stream velocity. This is done to isolate the prescribed motion effects from the rotor tilt effects. Second, both in translation and rotation cases, the turbine is closer to the top wall than to the bottom wall. Only the hub height is given for the translation configuration since the prescribed motion velocity is the same for every point in the rotor.

Two reference frames can be identified in Fig. 1 (b). The global reference frame is the frame $e_x$-$e_y$-$e_z$ which is centered at point $(0,0,0)$ and is fixed. $e_y$ follows the right-hand rule. The initial coordinates of the rotor center in this frame are $(0,0,z_{rc})$. The surge, sway and heave motions are, respectively, translations in the $e_x$, $e_y$ and $e_z$ directions and are defined relative to the global reference frame. The prescribed reference frame is the frame $e_{xp}$-$e_{yp}$-$e_{zp}$ which is centered at the tower bottom and translates with the turbine. $e_{yp}$ follows the right-hand rule. The roll, pitch and yaw motions are, respectively, the rotations about the $e_{xp}$, $e_{yp}$ and $e_{zp}$. All the aforementioned motions are defined in the floating foundation context and can be visualized in Fig. 2. The prescribed motions are of the form:

$$p = \pm A_{\mathrm{p}}\sin(\omega_{\mathrm{p}}t), \tag{2}$$

where $p$ is the corresponding prescribed motion (in length units for translation and angle units for rotations), $A_{\mathrm{p}}$ is the prescribed motion amplitude and $\omega_{\mathrm{p}} = 2\pi f_{\mathrm{p}}$ is the prescribed motion angular frequency. The prescribed motion capability was implemented in YALES2 in order to perform the simulations. Therefore, the validation is included in Appendix A.

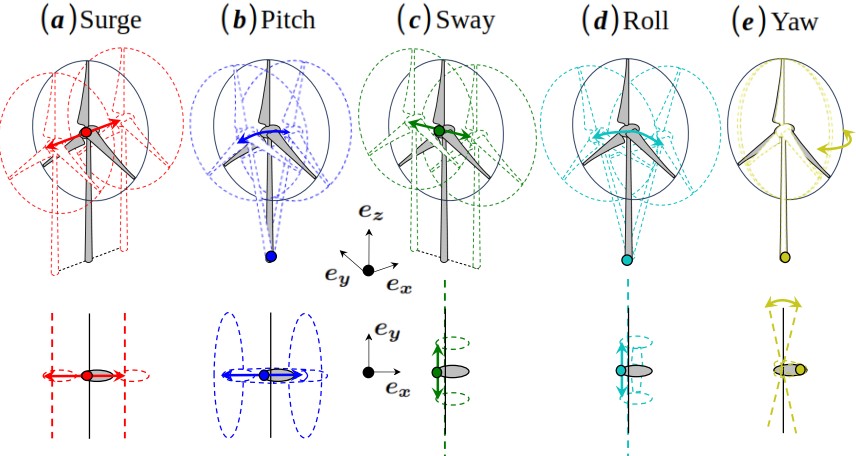

**Figure 2.** Floating foundation DOFs and the motion they induce on the turbine.

### 2.3 Case description

The turbine is modeled as rigid and operated near rated conditions, under constant rotor angular velocity $\omega_{\mathrm{r}}$ and steady uniform free-stream velocity of magnitude $U_0$. The operating conditions are summarized in Table 1. The simulated cases are described in Table 3 showing the amplitudes prescribed in each DOF $A_{\mathrm{p}}$, the corresponding non-dimensional values of the prescribed motion amplitude at the rotor center $A^*_{\mathrm{prc}}$, the prescribed motion frequency $f_{\mathrm{p}}$, and its non-dimensional value $St_{\mathrm{p}}$.



**Table 3.** Case definition.

| Case | DOF | $A_\mathrm{p}$ | $A_\mathrm{prc}^*$ $[-]$ | $f_\mathrm{p}$ $[Hz]$ | $St_\mathrm{p}$ $[-]$ |
|------|-----|------|------|------|------|
| FB | Fixed-bottom | 0 m | 0 | 0 | 0 |
| SuLS | Surge | 0.125 m | 0.0525 | 0.125 | 0.0744 |
| SuHS | Surge | 0.008 m | 0.0034 | 2.000 | 1.1905 |
| PiLS | Pitch | 3.0 deg | 0.0326 | 0.125 | 0.0744 |
| PiHS | Pitch | 0.3 deg | 0.0033 | 2.000 | 1.1905 |
| SPLS | Surge, Pitch | 0.125 m, 3.0 deg | 0.0851 | 0.125 | 0.0744 |
| SPHS | Surge, Pitch | 0.008 m, 0.3 deg | 0.0067 | 2.000 | 1.1905 |
| SwLS | Sway | 0.125 m | 0.0525 | 0.125 | 0.0744 |
| SwHS | Sway | 0.008 m | 0.0034 | 2.000 | 1.1905 |
| RoLS | Roll | 3.0 deg | 0.0321 | 0.125 | 0.0744 |
| RoHS | Roll | 0.3 deg | 0.0032 | 2.000 | 1.1905 |
| YaLS | Yaw | 3.0 deg | 0.0268 | 0.125 | 0.0744 |
| YaHS | Yaw | 0.3 deg | 0.0027 | 2.000 | 1.1905 |

The superscript $^*$ indicates that a quantity was normalized by the simulated laboratory-scale turbine diameter $D$. The non-dimensional frequencies are defined by the Strouhal number:

$$St_\mathrm{p} = \frac{f_\mathrm{p} D}{U_0}. \tag{3}$$

The cases are named with the two first letters of the DOF of the prescribed motion (e.g "Su" for surge), with the exception of the coupled surge-pitch motions that started with "SP". These letters are followed by either "LS" or "HS" standing for "Low Strouhal" or "High Strouhal", respectively. The dimensional values are scaled down from the real-scale DTU 10 MW turbine using the length and velocity scaling factors of $\lambda_L = 75$ and $\lambda_U = 3$, respectively. For the translational DOFs, $A_\mathrm{prc}^* = A_\mathrm{p}/D$. For rotational DOFs, $A_\mathrm{prc}^* = A_\mathrm{p} L_\mathrm{rp}/D$, where $L_\mathrm{rp}$ is the distance perpendicular to the rotation point (see Table A2 where $A_\mathrm{p} L_\mathrm{rp} = A_\mathrm{pb}$ for $R = 0$ m). For the coupled surge-pitch cases, the amplitudes are obtained by summing the amplitudes of the corresponding single-DOF surge and pitch cases, since the pitch amplitudes are small. All the amplitudes are defined relative to the initial turbine setup in Fig. 1 (b). The same frequency set is used for all DOFs. On the one hand, surge and sway are simulated with the same amplitude set. On the other hand, roll, pitch and yaw are also simulated with the same amplitude set. While the frequency-amplitude pairs are realistic for surge and pitch, the amplitudes for the remaining cases are over what one would normally expect for a FOWT, since the largest amplitudes tend to occur in the flow direction. Nevertheless, this allows for a comparison at similar magnitudes of prescribed motion among translation DOFs and among rotation DOFs, and it is possible to explore what phenomena could be triggered if one purportedly designed for larger sway, roll and yaw as a way to boost wake recovery. The validation of the prescribed motion implementation can be seen in Appendix A.





The rotor is placed in a computational domain of dimensions $19D \times 5.8D \times 1.6D$ and centered at a distance $5D$ from the domain inlet. The mesh is unstructured and composed of around 318 million tetrahedral elements. Figure 3 (a) and (b) show a longitudinal and transversal slice depicting the non-dimensional mesh resolution $D/\Delta x$, where $\Delta x$ is the cell size.

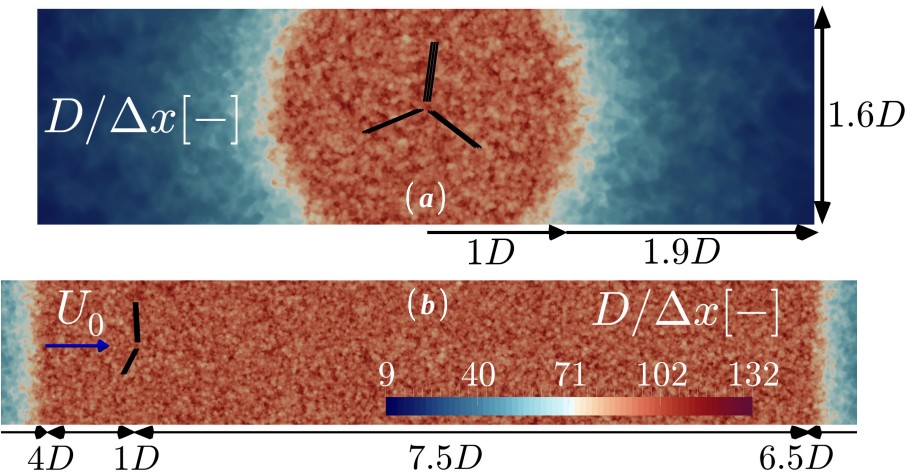

**Figure 3.** Longitudinal slice of the domain (a) and transversal slice of the domain (b) showing the normalized mesh resolution. The inlet and outlet are trimmed in (b).

This quantity is the inverse of the non-dimensional cell size and represents how many actuator points would exist in a rotor diameter, given the cell size of the point in question. Both figures put in evidence the different refinement regions and the domain dimensions. Two refinement levels are used. The wake refinement level is the high-refinement region with a maximum cell size in the vicinity of the rotor equal to $3.2 \times 10^{-2}$ m that yields 66 actuator points per rotor diameter. The cell size grows at a rate of $1.07$ from the wake refinement up to the background refinement at the domain boundaries that has a minimum resolution of nine actuator-points per diameter.

Prior to the result analysis, the simulation setup was validated extensively. The airfoils' $Re$ are found to be within $5\%$ of their actual $Re$ for the fixed-bottom case in the most loaded blade region (between $s \approx 30\,\%$ and $s \approx 97\,\%$, where $s$ is the spanwise position). The time convergence validation for the simulated time interval can be seen in Appendix B. Finally, a comparison with experimental results is presented in Appendix C. After these validation steps, the numerical setup was found to be suitable to investigate the wake of a FOWT.

## 3 Wake

### 3.1 Velocity deficit recovery

The wake evolution is a critical part of the analysis when it comes to wind turbines because they are almost always operated in a wind farm. This is especially true for floating turbines since the floating motion modulates the wake. Hence, it is important





to have an idea of how much momentum is transported downstream as well as the turbulence that the turbine adds to the wake, since both these factors impact power production, ultimate loads, fatigue loads and the wake characteristics. In this subsection, the normalized time-averaged streamwise flow velocity $<U_x>/U_0$ is first looked at in circular sections perpendicular to the wake to visualize case-specific details. $< \cdot >$ denotes the time average which is computed after convergence (see Appendix B). The wake is characterized by extracting the time series of the three flow field velocity components at several streamwise positions, by means of radial probes, as the ones shown in Fig. 4 and Fig. 5.

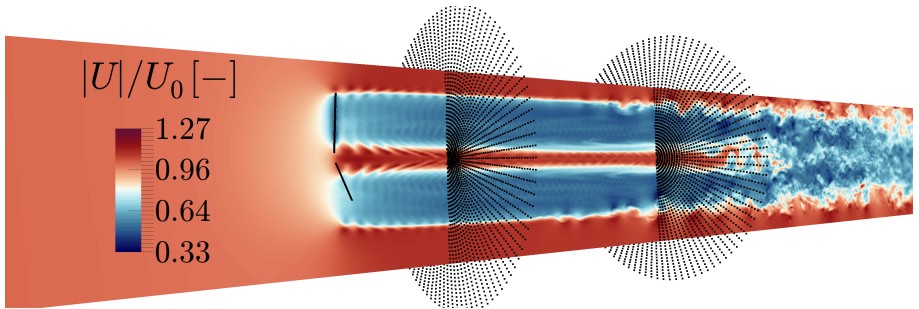

**Figure 4.** Two radial probes superimposed on a vertical domain slice. The slice is colored by streamwise velocity contours at $t/T_\mathrm{p} = 4$ for the SuLS case, where $T_\mathrm{p} = 1/f_\mathrm{p}$.

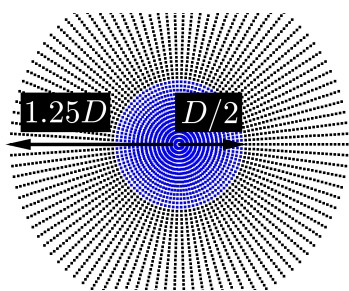

**Figure 5.** Radial probe representation. The blue circle delimits the rotor area.

Each radial probe is composed of 98 linear sub-probes dispersed azimuthally with a uniform angular spacing and discretized in 41 points in the radial direction. The number of sub-probes and their points is a compromise between spatial resolution and data size. The radial probe radius is $1.25D$ with a radial resolution of about $33/D$ and a minimum azimuthal resolution at the rotor perimeter of about $31/D$. The time series are acquired at each point in the probe. The probes' axes are aligned with the initial rotor axis throughout the simulation and do not move with the turbine. The blue circle in Fig. 5 delimits the rotor area, where the analysis is focused.

The streamwise evolution of $<U_x>/U_0$ can be seen in Fig. 6 for several cases. The black dashed circle indicates the rotor perimeter. Each row represents one downstream position while each column represents one case. The smooth plots are obtained via linear interpolation. Looking at the FB case, the time-averaged wake steadily expands downstream while conserving the





circular shape. This is a pattern that is observed for the low-$St$/high-$A^*$ and SuHS cases but not for the remaining high-$St$/low-$A^*$ cases (see Fig. D1). For the latter cases, the time-averaged wake develops more irregular and smeared-out edges than in the previous cases. These results suggest two things: (1) Exciting the wake at the high-$St$/low-$A^*$ leads to more destabilization at its outer boundary than with the low-$St$/high-$A^*$; (2) Exciting the wake at least in one direction perpendicular to the flow, i.e. all DOFs but surge, leads to more destabilization than exciting the wake exclusively in the flow direction, i.e. surge.

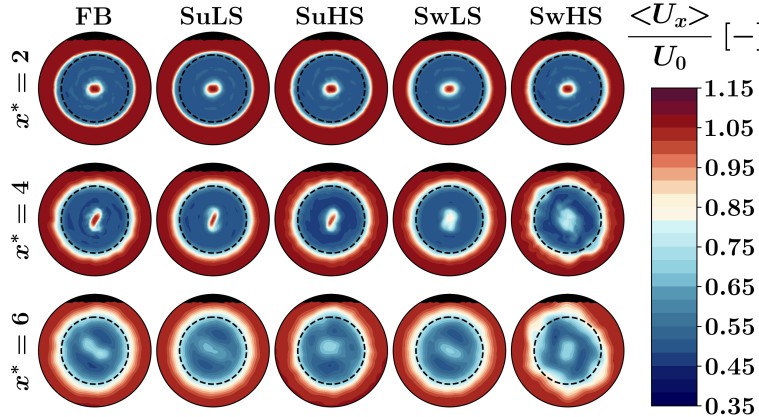

**Figure 6.** Normalized time-averaged streamwise velocity over the radial probes at several streamwise positions.

While the 2D circular sections are good to appreciate the spatial characteristics of the wake, they do not provide a practical way of comparing all the cases. In order to do this, the spatially averaged values of $<U_x>/U_0$ are computed over the area between the hub and the rotor radius for each radial probe which excludes the central jet that does not occur in reality. This quantity is simply called "wake recovery" throughout the paper. For a general variable $X$, the spatial averaging yields:

$$\overline{X} = \frac{\int_0^{2\pi} \int_{r_{\mathrm{h}}}^{r_{\mathrm{r}}} X \, r dr d\theta}{\int_0^{2\pi} \int_{r_{\mathrm{h}}}^{r_{\mathrm{r}}} \, r dr d\theta}, \tag{4}$$

where $\overline{\phantom{x}}$ denotes the spatial average, $r_{\mathrm{r}}$ is the rotor radius and $r_{\mathrm{h}}$ is the hub radius. The prescribed motion excess wake recovery can be computed by taking the difference in $\overline{<U_x>}/U_0$ relative to the FB case:

$$\Delta_{\mathrm{FB}} = \frac{\overline{<U_x>}}{U_0} - \frac{\overline{<U_x>}}{U_0}\bigg|_{\mathrm{FB}}. \tag{5}$$

In turn, the excess wake recovery speed is estimated through the slope of the previous equation normalized by $1/D$:

$$D\frac{\partial \Delta_{\mathrm{FB}}}{\partial x} = \frac{D}{U_0}\left(\frac{\partial \overline{<U_x>}}{\partial x} - \frac{\partial \overline{<U_x>}}{\partial x}\bigg|_{\mathrm{FB}}\right), \tag{6}$$

where the slope is calculated using forward finite differences. For each quantity, the result is a single number per streamwise position that is much easier to use for comparison. Figure 7 shows (a) $\overline{<U_x>}/U_0$, (b) $\Delta_{\mathrm{FB}}$ and (c) $D\partial\Delta_{\mathrm{FB}}/\partial x$ for several



streamwise positions. Figure 7 (a) illustrates the wake recovery profile for the FB case which is representative of the shape of the remaining cases' profiles. Although not shown, all cases show the minimum $\overline{<U_x>}/U_0$, i.e. the onset of the wake recovery, at $x^* = 3$, with the exception of cases SwHS and RoHS, whose minimum is located at $x^* = 2$. It should be kept in mind that there is some uncertainty since the distance between probes is $1D$.

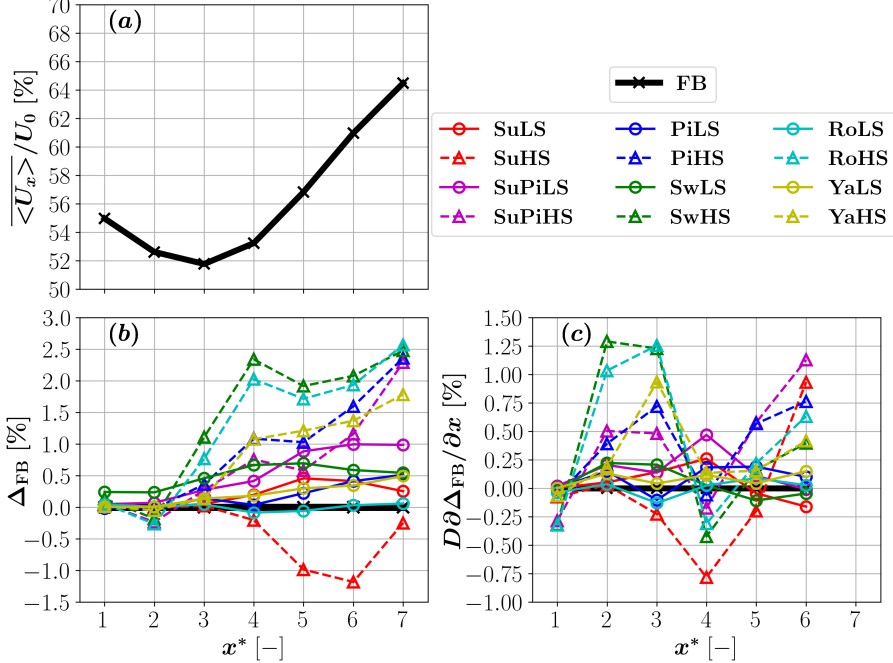

**Figure 7.** Streamwise evolution of the wake recovery for the FB case (a), prescribed motion excess wake recovery relative to the FB case (b) and prescribed motion excess wake recovery speed relative to the FB case (c).

Looking at Fig. 7 (b), the high-$St$/low-$A^*$ cases exhibit higher wake recovery than the low-$St$/high-$A^*$ cases, with the exception of surge that seems to hamper it between $x^* = 4$ and $x^* = 7$. The low-$St$/high-$A^*$ cases also lead to higher wake recovery when compared to the FB case, but the improvement is smaller. The recovery onset positions and excess recovery trends for the sway cases are similar to the closest simulated $St$ in Li et al. (2022). Despite this, the overall excess recovery values for the high-

$St$/low-$A^*$ cases are noticeably lower than those reported in Li et al. (2022) and Messmer et al. (2024) but one could argue that this is due to the fact that the simulated $St$ and associated $A^*$ values are not an exact match and are outside the most favorable range identified by the previous authors. Regarding surge, Messmer et al. (2024) reports a drop in the excess wake recovery in surge when the $St$ goes from 0.81 to 0.97. Given that the SuHS case was performed at $St = 1.1905$, it is not impossible to imagine after a certain $St$ threshold at certain values of $A^*$, that the wake recovery for this DOF could drop

below that of the FB case. This detrimental effect is also reported for some sway cases at $St = 0.1$ and $St = 0.8$ in Li et al. (2022). Moving back to the results, adding the surge motion to the pitch motion at high-$St$/low-$A^*$ has a detrimental effect on the recovery in all positions but the final value is very close to that of pitch. Adding the surge motion to the pitch motion at





low-$St$/high-$A^*$ increases the wake recovery at all positions. This can indicate that the effect of combining many motions will likely be a function of $St$ and $A^*$. Now turning to Fig. 7 (c), the SwHS, RoHS and YaHS cases present the maximum excess wake recovery speed between $x^* = 2$ and $x^* = 3$. Conversely, the SuHS, PiHS and SuPiHS present the maximum farther downstream at $x^* = 6$. This is another evidence of the difference between disturbing the flow perpendicularly to the flow versus in the direction of the flow. In the former, the recovery tends to accelerate more upstream and moderate after, while in the latter, the opposite occurs. This can be further corroborated by analyzing the PiHS, SuHS and SuPiHS cases. The PiHS case disturbs the flow both perpendicularly and in the direction of the flow, and has approximately the same excess wake recovery speed value at $x^* = 3$ and $x^* = 6$. When the surge motion is added (case SuPiHS), the excess wake recovery speed decreases at $x^* = 3$ and increases at $x^* = 6$, thus moving the maximum position farther downstream and closer to the surge trend. At low-$St$/high-$A^*$, there are no clear trends and the excess wake recovery speeds all seem to hover around the ones of the FB case, except for the surge-pitch case that shows a more pronounced peak at $x^* = 4$. The reason for the hampered recovery in SuHS could be explained by analyzing the rotor-centered vertical slices of normalized instantaneous velocity magnitude $|U|/U_0$ in Fig. 8, for the SuHS (a) and SwHS (b) cases.

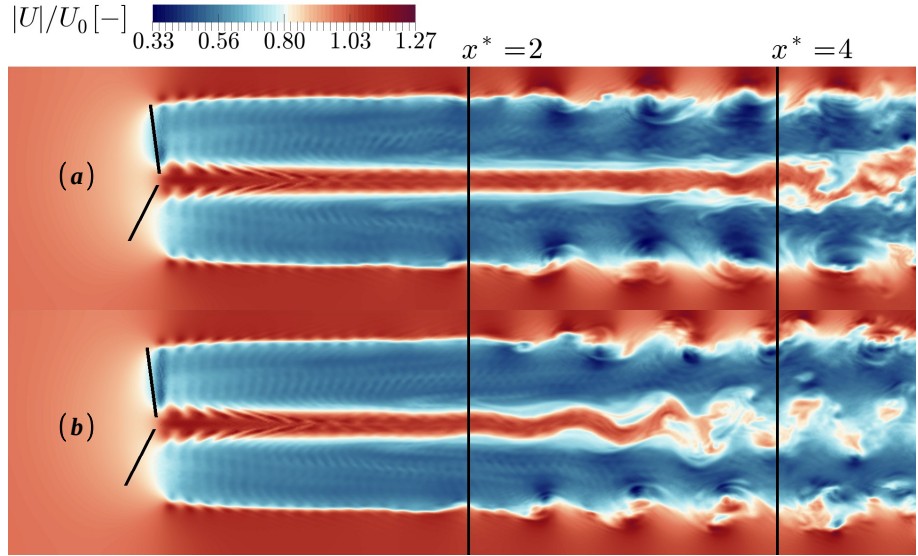

**Figure 8.** Normalized instantaneous velocity magnitude at the rotor-centered vertical slice for the SuHS (a) and SwHS (b) cases. The two black lines indicate streamwise positions.

The inner jet is less destabilized by the root vortices in surge between $x^* = 2$ and $x^* = 4$ than by the sway motion, a fact that hinders momentum entrainment in that region, leading to a slower recovery. The same observation was made for the rotor-centered horizontal slices (not shown). This difference highlights the importance of including the nacelle when performing the simulations, since the interior jet can then energize the wake in a way that may not happen in reality and can lead to differences in the wake evolution.





## 3.2 Turbulence intensity

The previously identified instantaneous wake instabilities may be present for all the cases, both at the root vortex and tip vortex regions. These instabilities are known to lead to changes in flow velocity in time and space that would be reflected in the turbulence intensity values over the wake. With the goal of measuring the prescribed motion impact on the turbulence intensity

in the three spatial directions, one can use the unidirectional turbulence intensity $TI_i$, defined as:

$$TI_i = \frac{<U_i'^2>^{1/2}}{<|U|>}, \tag{7}$$

where $U_i' = U_i - <U_i>$, $i \in \{x, y, z\}$ and $|\cdot|$ denotes the vector norm. The streamwise evolution of $TI_x$ can be seen in Fig. 9 for several cases. All floating cases show higher $TI_x$ values, both inside and outside the rotor area when compared to the FB case, at $x^* = 2$ (see Fig. D2 for the complete set of cases). The $TI_x$ spread tends to follow the prescribed motion (e.g, higher

side-to-side spread for sway). The spread is also observed in the low-$St$/high-$A^*$ cases but in a noticeably smaller area with higher concentration at the perimeter. Figure 9 further supports the observation of stronger wake destabilization in the high-$St$/low-$A^*$ cases, that is happening not only at the outer bounds but also in its interior. For the low-$St$/high-$A^*$ cases, the wake shows a well defined outer ring and inner core, at the tip and root vortices' positions, until $x^* = 4$, similarly to the FB case.

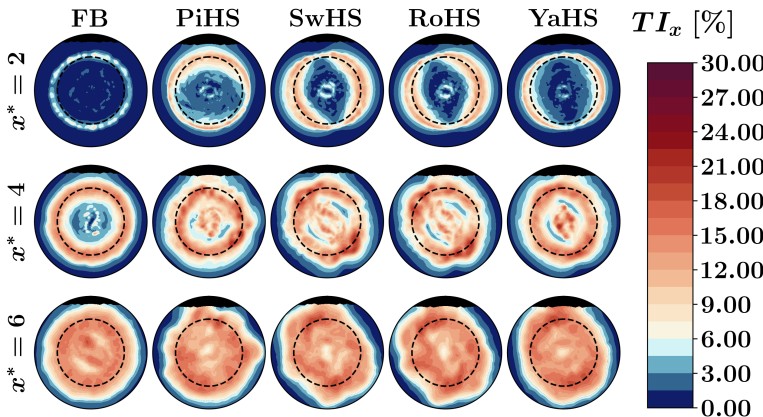

**Figure 9.** Streamwise turbulence intensity over the radial probes at several downstream positions.

Figure 10 (a) shows the spatially averaged values of both the streamwise and side-to-side turbulence intensities for the FB

case. This is done to show the general trend that the curves take, with both turbulence intensities ramping up along the stream and peaking around $x^* = 6$. Subplots (b) and (c) show the prescribed motion excess streamwise and side-to-side turbulence intensities between all cases and the FB case, respectively. The excess turbulence intensities are calculated with an expression analogous to Eq. (5). The high-$St$/low-$A^*$ cases show higher $\overline{TI_x}$ values than the low-$St$/high-$A^*$ between $x^* = 2$ and $x^* = 4$ and vice-versa after $x^* = 5$. The high-$St$/low-$A^*$ values diverge upwards and converge downwards very quickly towards the





FB case values, while the low-$St$/high-$A^*$ values maintain an almost constant difference until $x^* = 5$, before dropping. For $\overline{TI_y}$, the high-$St$/low-$A^*$ cases show higher values than the low-$St$/high-$A^*$ from $x^* = 2$ onward, with peaks between $x^* = 3$ and $x^* = 4$. The trend for the low-$St$/high-$A^*$ cases is close to that of the fixed-bottom case for this component, with the exception of the coupled surge-pitch case that shows an increasing difference up until $x^* = 5$.

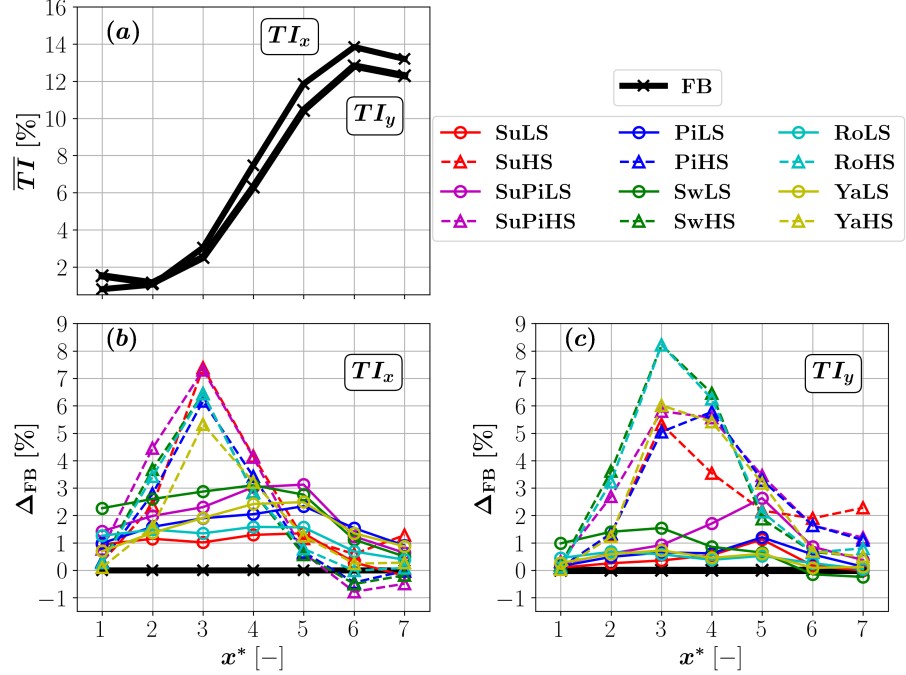

**Figure 10.** (a) Streamwise evolution of the spatially averaged turbulence intensity in two directions for the FB case; (b) Prescribed motion excess streamwise turbulence intensity between all cases and the FB case; (c) Prescribed motion excess side-to-side turbulence intensity between all cases and the FB case.

Similar trends are found for sway by Li et al. (2022) for the closest simulated $St$ numbers, namely a more upstream $TI$ increase
for the highest $St$ numbers. The results show a link between higher values of $\Delta_{\mathrm{FB}}$ of $\overline{TI_y}$ (and $\overline{TI_z}$, although not shown) and a faster wake recovery that is found by comparing all the high-$St$/low-$A^*$ cases to the low-$St$/high-$A^*$ and FB cases (with the exception of SuHS, although $\overline{TI_y}$ drops more upstream for this case). The link makes sense because there is more momentum available for entrainment in directions perpendicular to the flow, i.e. outside the wake, than in the direction of the flow itself, i.e. in the wake. Adding the pitch motion to the surge motion at high-$St$/low-$A^*$ (case SuPiHS), appears to lead to some $TI$
summation before $x^* = 3$. After $x^* = 5$, the coupled motion $\overline{TI}$ curves are close to those of pitch, as if the dynamics is driven by this DOF. The trends of the corresponding excess wake recovery values are close between the coupled motion and pitch too, although with an offset from $x^* = 3$ to $x^* = 7$. For the low-$St$/high-$A^*$ cases, the coupling seems to have some degree of coupled effect on the wake, in that the $\overline{TI}$ values of the coupled motion are higher than surge and pitch alone, with the same happening for the recovery.





### 3.3 Frequency content

So far, it is possible to link the directionality and the frequency-amplitude pair of the prescribed motion to the wake recovery and turbulence intensity values observed in the wake. However, this is not direct evidence that the prescribed motion is indeed being transmitted to the wake and modulating its behavior. This evidence can be found by looking, for instance, at the side-to-side velocity component of the wake in the frequency domain and checking if the prescribed motion frequency is one of the strongest peaks, by means of the normalized power spectrum defined as:

$$\frac{P_{yy}(St)}{U_0{}^2} = \frac{|\widehat{U_y}(t)|^2}{U_0{}^2},\tag{8}$$

where $\widehat{\ }$ denotes the fast Fourier transform (FFT), $|\cdot|$ denotes the complex number modulus and the frequency $f$ was non-dimensionalized into $St$. For each radial probe, the results are spatially averaged over each probe point inside the rotor perimeter with a simple sample average, leading to the spatially averaged wake spectrum, defined by:

$$\frac{\overline{P_{yy}}(St)}{U_0^2} = \frac{1}{N} \sum_{n=1}^{N} \frac{P_{yy}(St)}{U_0{}^2},\tag{9}$$

where $N$ is the number of radial probe points inside the rotor perimeter. The sample average is used in order to decrease the computational expense of the post-processing and only because the goal was to identify the main frequencies in the wake, for which the sample average is enough.

Figure 11 shows the streamwise evolution of this quantity for the FB and both sway cases. The plot indicates that the FB case develops a high-energy region for approximately $1 \leq St \leq 2$ at $x^* = 2$ and in the region surrounding $St_r$. These oscillations moderate relative to the rest of the spectrum and concentrate more in the range of approximately $0.5 \leq St \leq 1.5$ at $x^* = 4$. At $x^* = 6$, only some reminiscent peaks are found in the range $0.5 \leq St \leq 1$ with a peak at $St = 0.5$. The slope at high frequencies approaches minus five-thirds, as expected for free decaying turbulence at high $Re$. The identified peak frequency $St = 0.5$ at $x^* = 6$ and $x^* = 7$ is close to those found in the literature (Heisel et al., 2018) and they appear to shift to lower frequencies as the wake progresses downstream (see Fig. D3 for the remaining downstream positions). Both the SwLf and SwHf cases show peaks at $St_{\mathrm{p}}$, $St_{\mathrm{r}}$ and their multiples at $x^* = 2$. At the same position, both cases also show peaks at the sum and differences between $St_{\mathrm{r}}$ and $St_{\mathrm{p}}$, and between their multiples. This leads to a high-energy region in SwLf around $St = 3f_{\mathrm{r}}D/U_0 = 7.14$. This is explained by the fact that, since $St_{\mathrm{p}} \ll 3St_{\mathrm{r}}$, then $3St_{\mathrm{r}} - St_{\mathrm{p}}$ is close to $3St_{\mathrm{r}}$ and therefore, the sums and differences are concentrated around $3St_{\mathrm{r}}$. For SwHf, well-defined sums and differences are present since $St_{\mathrm{p}} \sim 3St_{\mathrm{r}}$. For both prescribed motion cases, the prescribed motion frequency is amplified between $x^* = 1$ and $x^* = 4$ and decays afterwards, as the wake progresses downstream (see Fig. D3 for the rest of the results). At $x^* = 6$, only the prescribed motion frequency shows a strong peak for SwHS, while for SwLS the prescribed motion frequency peak is of the same order as the frequencies of the previously identified high-energy region in the FB case. Until $x^* = 5$, the highest peaks for both simulation groups coincide with the prescribed motion frequency (see Fig. D3), in accordance with Mao and Sørensen (2018). The power in all spectra increases significantly from $x^* = 1$ to $x^* = 4$, while from $x^* = 4$ to $x^* = 7$ there is stabilization and convergence towards the free decaying turbulence spectrum.



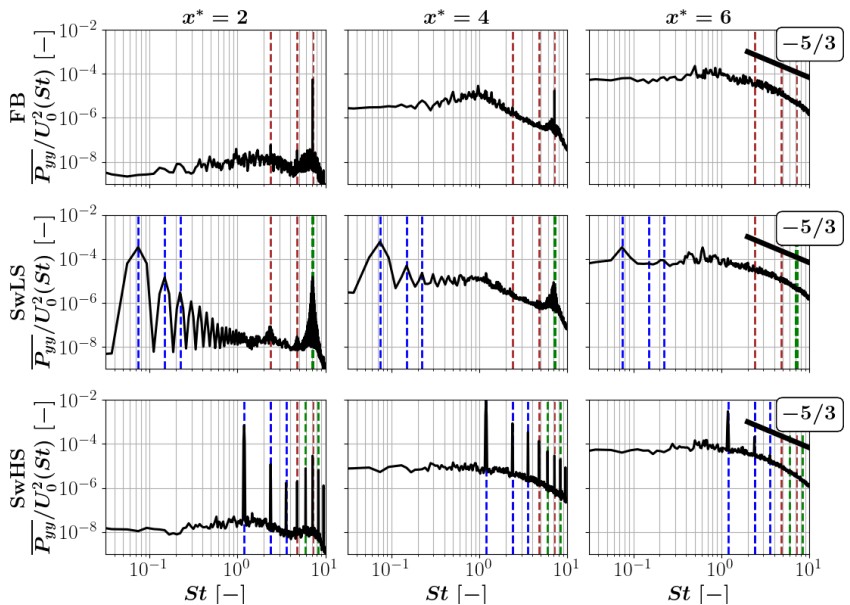

**Figure 11.** Spatially averaged wake spectra of the side-to-side velocity component. The red lines are the first three multiples of the rotational frequency $St_{\mathrm{r}}$. The blue lines are the first three multiples of the prescribed motion frequency $St_{\mathrm{p}}$. The green lines are the sum and difference between $3St_{\mathrm{r}}$ and $St_{\mathrm{p}}$.

The figure suggests an explanation as to why the high-$St$/low-$A^*$ cases present stronger wake perturbations (indicated by higher $TI$ values) than the low-$St$/high-$A^*$ cases. This could be happening because, in the former cases, the prescribed frequency is within the frequency range at which the wake develops velocity oscillations when no motion is prescribed (FB case), while this is not true for the latter cases. As a final observation, and although not included in this paper, the spectra of the remaining high-$St$/low-$A^*$ and low-$St$/high-$A^*$ cases are very similar to the spectra of SwHS and SwLS, respectively.

The previously shown spectra give a general overview of the evolution of characteristic wake frequencies, both with and without prescribed motion (i.e., upstream perturbation). Focusing now on the prescribed motion cases, it is relevant to assess how much the upstream perturbations are amplified as the wake progresses. For that, one can use the normalized velocity amplitude spectrum at the prescribed motion frequency to assess the velocity perturbations against the free-stream velocity, which is defined by:

$$\frac{A_i(St_{\mathrm{p}})}{U_0} = \left. \frac{|\widehat{U_i}(t)|}{U_0} \right|_{St=St_{\mathrm{p}}}, \tag{10}$$

where $i$ denotes the considered direction. On the other hand, it is paramount to assess the velocity perturbations against the initial perturbation, as some prescribed motions may have a stronger presence in the wake simply because the initial perturbation is stronger. For that, amplification factors are used to quantify the ratio of both and are written as follows:

$$k_i(St_{\mathrm{p}}) = \frac{\overline{A_i}(St_{\mathrm{p}})}{A_{\mathrm{prc}}\omega_{\mathrm{p}}}, \tag{11}$$





where $\overline{A_i}(St_\mathrm{p})$ is the spatially averaged value of $A_i(St_\mathrm{p})$ using an expression analogous to Eq. (4), $A_\mathrm{prc}$ is the prescribed motion amplitude at the rotor center and $\omega_\mathrm{p}$ is the prescribed motion angular frequency. In essence, $A_\mathrm{prc}\omega_\mathrm{p}$ represents the prescribed motion velocity amplitude at the rotor center. For yaw, the maximum perturbation velocity is computed at the tip

of a blade with an azimuth of $\theta = 90$ deg, assuming a non-rotating rotor. It makes sense to do this because the magnitude of this value is the same as if the rotor center velocity was computed at a reference frame fixed at the tip of the blade. Figure 12 shows the spatially averaged normalized side-to-side velocity amplitude spectrum at the prescribed motion frequency (a), and the corresponding amplification factors (b).

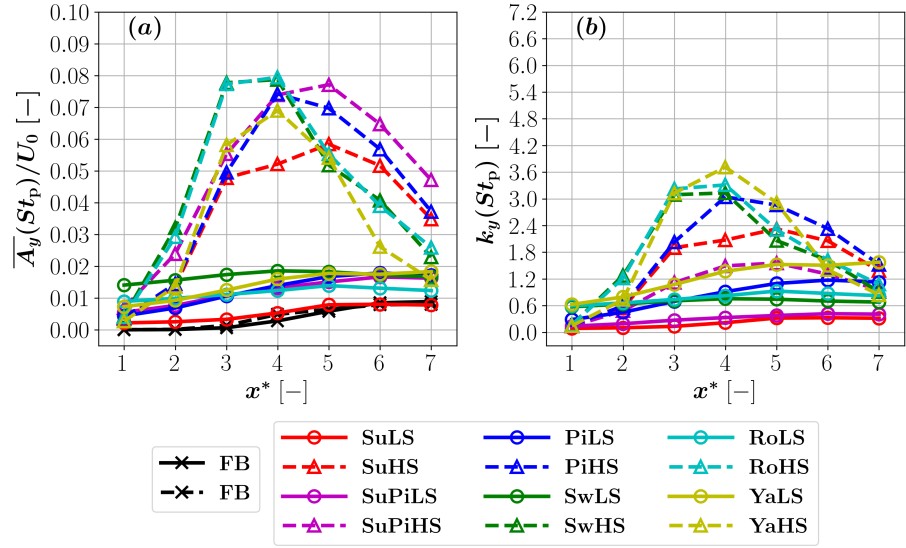

**Figure 12.** (a) Spatially averaged normalized side-to-side velocity amplitude spectrum at the prescribed motion frequency; (b) side-to-side amplification factors at the prescribed motion frequency.

The analysis of $\overline{A_y}(St_\mathrm{p})/U_0$ demonstrates that the prescribed motion is indeed modulating the wake behavior. All the high-

$St$/low-$A^*$ cases show a sharp increase in $\overline{A_y}(St_\mathrm{p})/U_0$ in the region $2 \leq x^* \leq 3$ that peaks in $3 \leq x^* \leq 5$ and decays for larger values of $x^*$, which is generally consistent with the highest simulated $St$ in Li et al. (2022) and Messmer et al. (2024). The cases with more perturbations perpendicular to the main flow direction peak at smaller $x^*$ than the cases with more flow-aligned perturbations. The low-$St$/high-$A^*$ cases also show an increase in $\overline{A_y}(St_\mathrm{p})/U_0$ but much lower, farther downstream than the previous cases and with subsequent stabilization, similarly to the results for $St = 0.2$ in Li et al. (2022). The low-$St$/high-$A^*$

surge case (SuLf) is exceptional, in that it does not show any meaningful increase at all when compared to the FB case. Coupling pitch to surge has very little effect, given that the curves remain very similar to those of pitch, highlighting once again its dominance. At high-$St$/low-$A^*$ though, the peak shifts farther downstream, closer to the surge peak position. Looking at the amplification factor $k_y(St_\mathrm{p})$ for a fairer comparison among cases, the high-$St$/low-$A^*$ cases generally exhibit higher amplification than the low-$St$/high-$A^*$ cases. The surge-pitch coupling is an outlier since summing the surge and pitch perturbations





leads to a disproportionately small increase in the spectral component (see Fig. 12 (a)). That is why the amplification factor of the coupled case is lower than those of the single DOFs alone. The high-$St$/low-$A^*$ yaw case (YaHf) is the most amplified of all cases, suggesting that its directionality is more prone to wake disturbances, followed by roll, sway and pitch. The surge cases present the lowest amplification of the single DOFs showing that wake perturbations in the direction of the flow have a limited development potential into perturbations perpendicular to the flow. As for low-$St$/high-$A^*$, similar conclusions as

before can be drawn, but with the amplification factors steadily increasing as the wake progressed downstream.

     Up until now, a strong link has been identified between the prescribed motion and the wake disturbances. However, the transmission mechanisms and the effects on the wake physics are still elusive. With the goal of identifying the underlying physics, one can observe how $A_y(St_\mathrm{p})/U_0$ is distributed across each radial probe. This is shown along the columns of Fig. 13 for the surge and sway cases, and along the rows for some streamwise positions. The white dashed circle indicates the rotor perimeter.

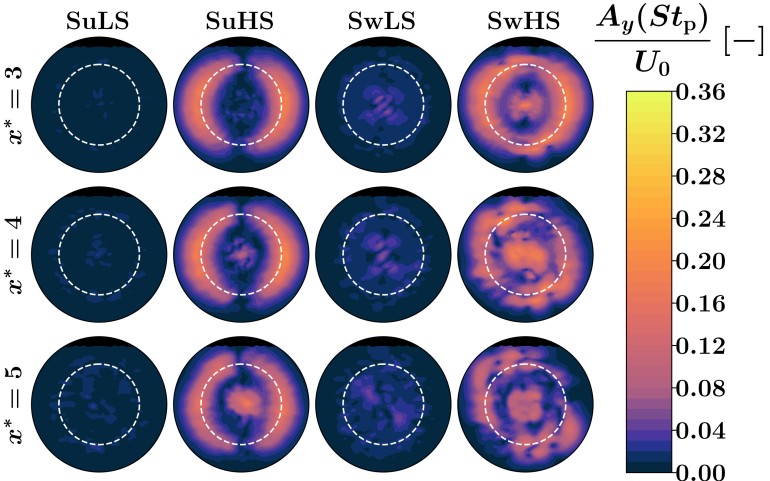

**Figure 13.** Normalized amplitude spectrum component of the side-to-side velocity at the prescribed motion frequency over the radial probes for the surge and sway cases.


Clear differences are identified between the high-$St$/low-$A^*$ and low-$St$/high-$A^*$ cases. In the former, and similarly to what was concluded before, the prescribed motion induces flow-perpendicular velocity oscillations while in the latter this does not happen, despite the fact that both groups converted the prescribed motion into flow-aligned velocity oscillations (see Fig. D4). This is probably the reason why the former group develops higher values of $TI_y$ and a faster recovery (see Fig. 10 and 7).

Moreover, when transmission occurs, it does so at the rotor perimeter and rotor center. However, in the rotor center, this occurs more upstream for the SwHS case and more downstream for the SuHS which is consistent with Fig. 8 and could explain the difference in the recovery speeds of Fig. 7.





## 3.4 Q-criterion

All in all, the transmission mechanisms appear to be located at the rotor diameter and near the rotor center which naturally
points towards the tip and root vortices and the associated shear layers. The tip vortices can easily be visualized by plotting the
normalized $Q$-criterion $QD^2/U_0^2$ that indicates regions where the local rotation is larger than the local strain rate. Figure 14
shows the surface $QD^2/U_0^2 = 2.48$ for the FB (a), SuLS (b), SuHS (c) and SwHS (d) cases, respectively. The SuLS case shows
virtually no difference with regards to the FB case, which is in accordance to all the previously identified wake similarities
and Messmer et al. (2024). The SuHS case, on the other hand, shows a pulsating tip vortex with a larger diameter than the
former case which is likely caused by the earlier onset of the leap-frogging phenomenon followed by the merger between two
consecutive tip vortices. These coherent structures are more pronounced and long-lived than in the former cases.

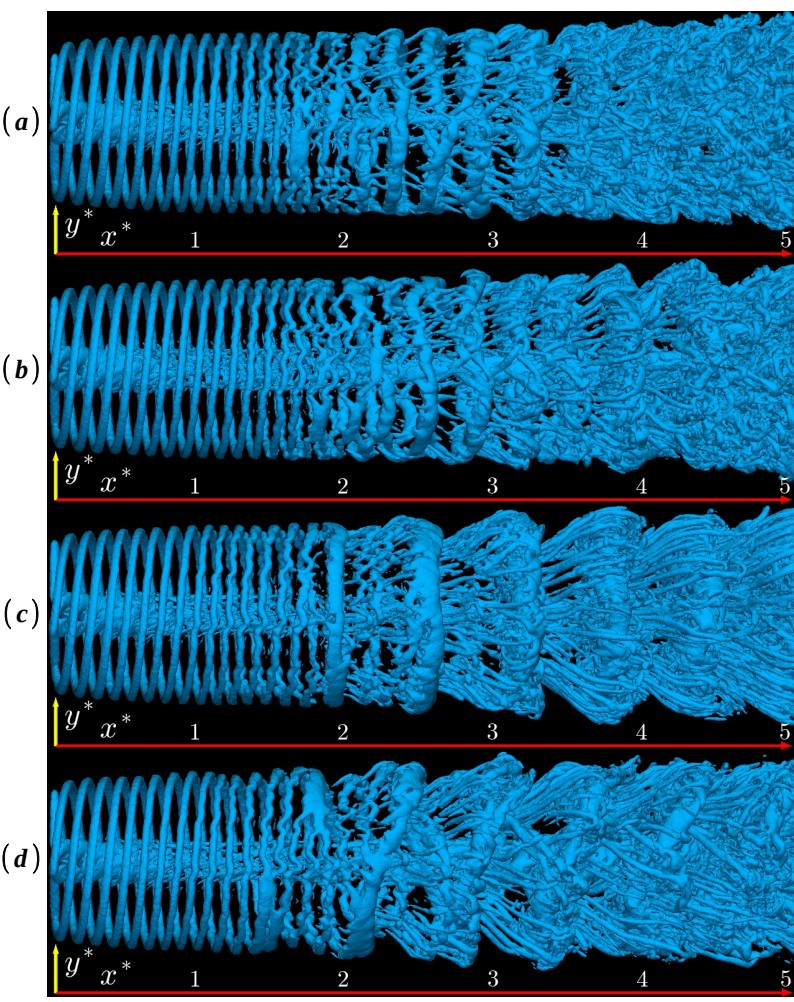

**Figure 14.** Normalized $Q$-criterion surface $QD^2/U_0^2 = 2.48$ for the FB (a), SuLS (b), SuHS (c), SwHS (d) case.





This type of behavior also reinforces the idea that the prescribed motion is inducing a resonance phenomenon, whereby the initial perturbation is amplified by the wake. Finally, in addition to what is observed for SuHS, the SwHS case depicts ripples in the upstream tip vortex envelope and a pulsating vortex that is diagonally deformed. This is consistent with the sway motion directionality and appears to lead to earlier breakdown of the tip vortices with less long-lived structures. While the $Q$-criterion provides visual proof of the effect of the prescribed motion, quantitative insights cannot be drawn from these visualizations. Instead, the tip and root vortices should be analyzed by looking at the corresponding shear layers in detail, which is done in the next section.

## 3.5 Shear layer

The shear layer is a key aspect of the wake recovery because it isolates the wake from the free flow, hindering momentum entrainment and slowing the recovery down. This means that accelerating the shear layer breakdown could speed up the recovery, which may explain the differences observed between the high-$St$/low-$A^*$ and low-$St$/high-$A^*$ cases. This hypothesis is investigated in detail at the domain slice defined by the plane $z^* = z_{\mathrm{rc}}^*$ in Fig. 1 (b), where $z_{\mathrm{rc}}^*$ is the normalized rotor center height. The shear layer is tracked in these planes using a normalized flow velocity standard deviation threshold on the velocity magnitude $<|U|'^2>^{1/2}/U_0$, above which the flow is considered to be inside the shear layer. For the purpose of this analysis, a more accurate definition is not required.

Figure 15 shows $<|U|'^2>^{1/2}/U_0$ for the SwLS (a) and SwHS (b) cases, respectively, together with the contour line of $<|U|'^2>^{1/2}/U_0 = 0.044$ in yellow.

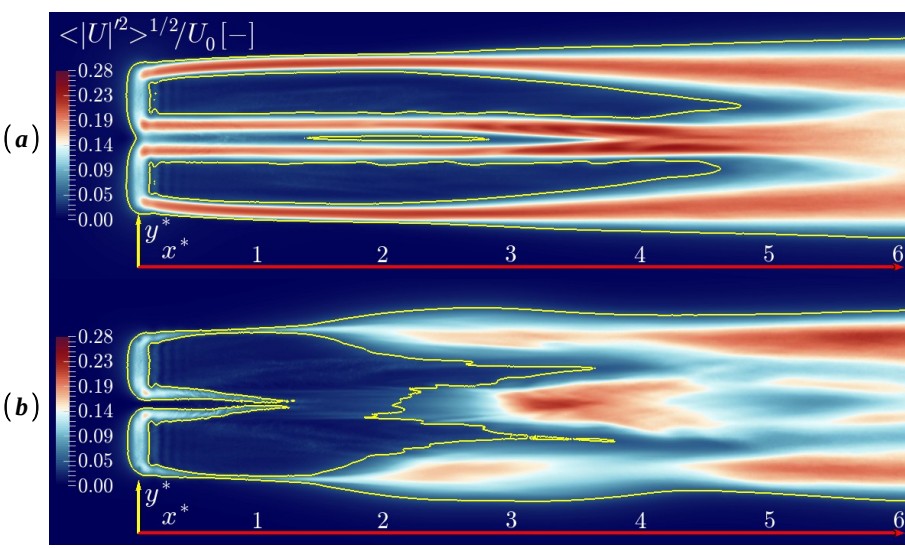

**Figure 15.** Normalized standard deviation of the velocity magnitude $<|U|'^2>^{1/2}/U_0$ for: (a) the SwLS case, and (b) the SwHS case. The plane is defined by the equation $z^* = z_{\mathrm{rc}}^*$ (see Fig. 1). The reference frame indicates the directions but not the origin.





Both contours show two regions of high $<|U|'^2>^{1/2}/U_0$ that emanate from the blade tips and roots and that are consistent
with what is expected from the tip and root vortices. However, the regions are somewhat different between both cases. In the
former, the tip and root shear layers seem to expand monotonically and remain almost unchanged, respectively. In the latter,
the tip shear layers appear to expand, contract and expand again, leading to a bulb-shaped contour, while the root shear layers
appear to contract and then expand.

Figure 16 shows the tip shear layers from Fig. 15 at the $y^* > y_{rc}^*$ region together with that of the fixed-bottom case FB (a)
and the tip shear layer thickness for all the cases (b). $y_{rc}^*$ is the normalized $y^*$ coordinate of the rotor center. It is evident that
the shear layers of the high-$St$/low-$A^*$ cases grow at a much faster rate than the ones from the low-$St$/high-$A^*$ and FB cases,
until $x^* = 3$. Afterwards, the former group's shear layers appear to contract until between $4 \leq x^* \leq 5$ and are followed by an
expansion at an accelerating rate, except for the SuHS case. The latter group's shear layers grow monotonically but the growth
rate sharply decreases at around $x^* = 5$. Close inspection of those cases reveals that the change in growth rate coincides with
the merger between the tip shear layer and the corresponding root shear layer. When this happens, the tip shear layer can no
longer expand inwards, hence the decrease in growth rate (see the lower boundary for case FB in subplot (a)). Compared to
the low-$St$/high-$A^*$ cases, the merger for the high-$St$/low-$A^*$ happens between $0.5D$ and $1.5D$ before, and the shear layer
expansion still manages to accelerate.

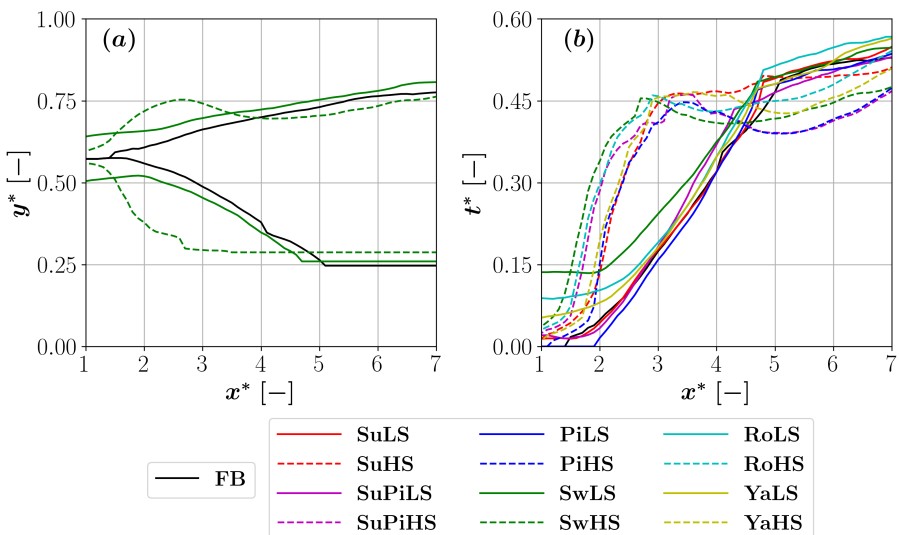

**Figure 16.** (a) Tip shear layer at the $y^* > y_{rc}^*$ region for the FB, SwLS and SwHS cases; (b) Shear layer thickness for all cases.

One can now compute the normalized time-averaged streamwise flow velocity that is spatially averaged over the shear layer
$\overline{<U>}/U_0$ - i.e., between the upper and lower boundaries of Fig. 16 (a) - and use it as proxy for momentum entrainment in that
region. This quantity can be seen in Fig. 17 in blue, together with the shear layer thickness in red, both as a function of the
downstream distance from the rotor. The full line refers to the region $y^* > y_{rc}^*$ and the dashed line refers to the region $y^* < y_{rc}^*$.



The cases are identified in bold in the subplots. The red vertical lines indicate the streamwise position at which the tip shear layer merges with the corresponding root shear layer.

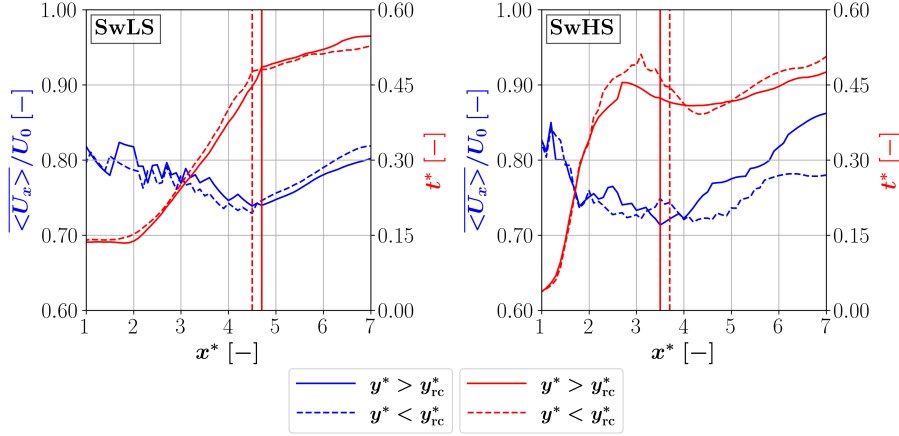

**Figure 17.** Normalized time-averaged streamwise flow velocity spatially averaged over the shear layer and tip shear layer thickness for the SwLS and SwHS cases in the plane $z^* = z^*_{\mathrm{rc}}$ (see Fig. 1).

The figure indicates that $\overline{<U>}/U_0$ decreases until and reaches a minimum at the point where the tip and root shear layers merge. From that point onwards, $\overline{<U>}/U_0$ starts to recover. In particular, the SwHS case is characterized by earlier mergers, which are consistent with a faster expansion and the earlier onset of the recovery. As mentioned in the subsection "Velocity deficit recovery", the onset of the wake recovery for all the cases happens at $x^* = 3$, with the exception of cases SwHS and RoHS, whose minimum is located at $x^* = 2$. This precedes the shear layer merger by a distance of about $1D$ for the majority

of the cases. However, according to the literature (Lignarolo et al., 2015; Messmer et al., 2024), the recovery is supposed to start at approximately the merger position. This difference could be explained by the uncertainty in determining the recovery onset due to the large radial probe spacing and by the fact that only a 2D slice of the shear layer is analyzed, which may skew the conclusions. Nevertheless, if the conclusions are not skewed, the recovery may have indeed started earlier since there is a high-momentum jet at the wake core due to the absence of the nacelle. The explanation for the merger-recovery coincidence

is also intuitive. While the tip shear layer is expanding, it expands both outwards towards a higher momentum region but also inwards towards a lower momentum region emanating from the blades (see Fig. 15 (b)). When the merger happens, the shear layer is no longer capable of expanding inwards and only expands outwards, leading to the recovery. As previously observed, the shear layers slightly contract in SwHS, keeping their thickness below those of SwLS but then the growth rate quickly picks up. Under the criterion used to track the shear layer boundary, the tip and root shear layers merge between $x^* = 3$ and $x^* = 5$

for all cases, which coincides with the range of steep increase in $TI$ (see Fig. 10) and is broadly in agreement with Kang et al. (2014). In SwHS, although both shear layers have approximately the same thickness, they have slightly disparate recoveries, with a difference of the order of $8\%$ of the free-stream velocity. This is a fact also observed for high-$St$/low-$A^*$ pitch, surge-pitch, roll and yaw cases, despite not being shown. Contrary to surge, all these motions deform the wake perpendicularly to

 

the flow, leading to a wake whose borders are not circular but rather elliptical (deformed in the directions of the motions)
and irregular. This combined with the fact that the turbine is closer to the top wall than to the bottom wall and that the wake
is rotating anti-clockwise could produce the observed differences. Since all lateral boundaries are slip walls, this means that
the more constrained (and arguably faster) flow near the top wall is transported to the semi-plane $y^* > y^*_{rc}$ and that the less
constrained (and arguably slower) flow near the bottom wall is transported to the semi-plane $y^* < y^*_{rc}$. This insight can explain
the differences in momentum observed between both tip shear layers. Although this difference would potentially not be present,
were the domain symmetrical and the turbine centered in the domain, it is still relevant to observe it because real floating turbine
wakes are effectively bounded by the water on one side and may, therefore, present this particularity. The same analysis was
performed at the rotor-centered vertical slice (see Fig. 8) but it is only provided in the appendix's Fig. D5. The reason for this is
that the shear layer near the top wall interacts with it relatively upstream, thus skewing the results. More specifically, after the
tip and root shear layer merger near the centerline, the shear layer growth stops upon the interaction with the top boundary. This
severely delays its recovery due to the absence of high-momentum flow in its vicinity, leading to less representative results.

Lastly, Fig. 18 plots the normalized side-to-side velocity amplitude spectrum at the prescribed motion frequency that is
spatially averaged over the shear layer $\overline{A_y}(St_p)/U_0$, together with the shear layer thickness.

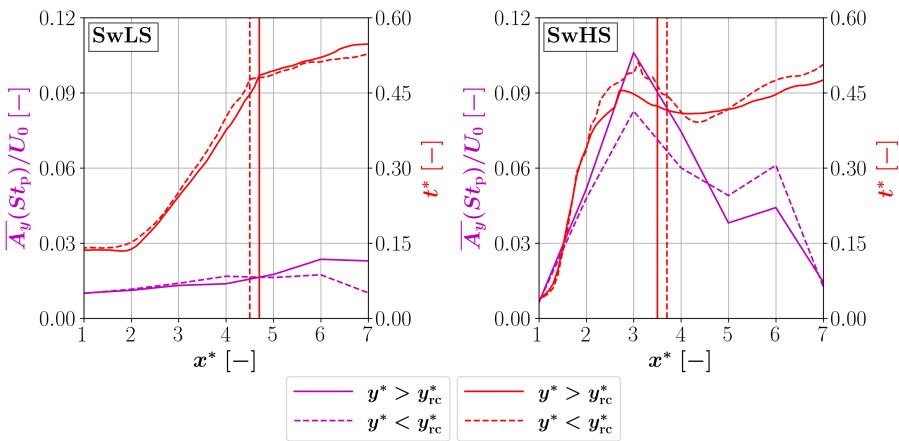

**Figure 18.** Normalized side-to-side velocity amplitude spectrum at the prescribed motion frequency spatially averaged over the shear layer
and tip shear layer thickness for the SwLS and SwHS cases in the plane $z^* = z^*_{rc}$ (see Fig. 1).

For SwLS, it can be seen that the shear layer expansion is not accompanied by an increase in $\overline{A_y}(St_p)/U_0$. Conversely, for
SwHS, there is a sharp increase in $\overline{A_y}(St_p)/U_0$ that peaks approximately at the maximum thickness position, providing strong
evidence that the early expansion is triggered by the prescribed motion. As previously noted by Mao and Sørensen (2018), this
figure illustrates the role that the shear layer has in amplifying upstream perturbations when the amplitude and frequency are
favorable, adding to the considerations associated with Fig. 13 and the subsection "Q-criterion".



## 3.6 Wake meandering

Another aspect of the wake that may be relevant for the recovery and downstream turbine load calculation is the wake mean-
dering that is induced by the prescribed motion. The wake meandering is quantified in this section through the estimation of the
instantaneous wake center. At every time step, a Gaussian profile is fit on the streamwise velocity component at each azimuth
position of each radial probe. An example of the Gaussian fit (GF) can be seen in Fig. 19.

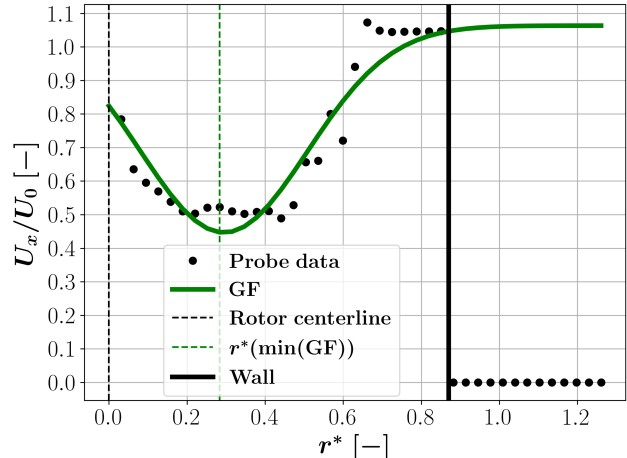

**Figure 19.** Radial probe streamwise velocity at $x^* = 5$ and $\theta = -33$ deg for case SwLS.

The fit is performed by neglecting the flow velocity values outside the domain and instead replacing them with the last value
inside the domain. This is done to ensure that the fit is performed across the same radial distance for all directions, both fully
or partially contained in the domain. The fact that the velocity minima are observed away from the centerline is due to the
absence of the nacelle in the simulations. This constitutes a strong argument in favor of performing the fit for each azimuth
independently, because fitting the whole wake or diametrically opposed azimuth angles altogether, would have required fitting
two Gaussian profiles instead of one. Besides this, fitting azimuthally constrains the fit less, allowing to better capture eventual
wake meandering and deformation. Following the fit, the instantaneous center coordinates are calculated as the average between
the fit minima over all the azimuth angles:

$$x^*_{i\,\text{wc}} = \frac{1}{98} \sum_{n=1}^{98} x^*_i(\min(\text{GF}(\theta_n))), \tag{12}$$

where $i$ denotes the considered direction and $\theta_n$ is the azimuth angle of the $n^{th}$ probe. The time series of the center coordinate
estimates can be seen in Fig. 20, where $T_\text{p} = 1/f_\text{p}$. The amplitude in the $y$ direction is much larger than that in the $z$ direction
but no considerations can be drawn from this fact since the wake is rotating. Finally, both coordinates are oscillating periodically
at what seems to be the prescribed motion frequency.

The motion can also be visualized by plotting one coordinate against the other at each instant in time. If the points are
then binned, one obtains the relative frequency distribution $f$ of the wake center, which is a proxy for the probability density





distribution of finding the wake center at a certain position. Figure 21 shows the relative frequency distribution for the FB, SwLS and SwHS cases in the first, second and third rows. Columns from left to right correspond to a sequence of streamwise

positions. The reference frame is centered at the coordinates $(y_{\mathrm{rc}}^*, z_{\mathrm{rc}}^*)$. The FB case displays a wake center highly concentrated at the center of the reference frame that starts to smear out from $x^* = 5$. The SwLS case, on the other hand, exhibits a highly concentrated side-to-side motion over a horizontal line at $x^* = 1$.

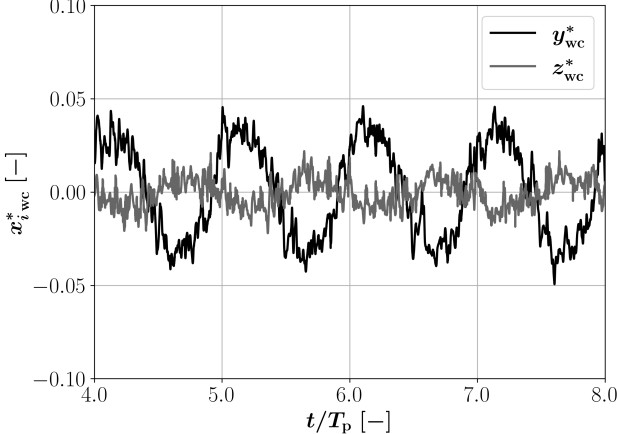

**Figure 20.** Instantaneous wake center coordinates at $x^* = 5$ for case SwLS.

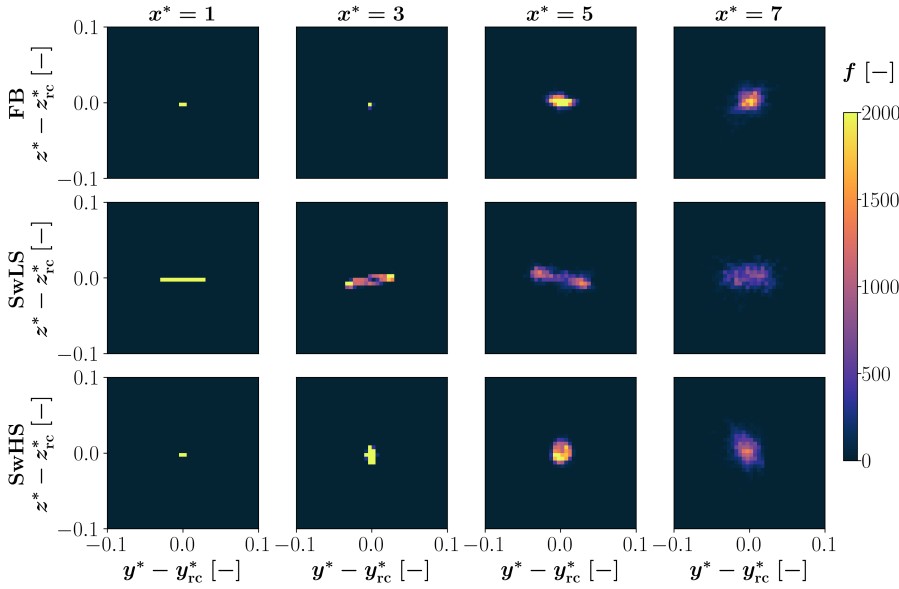

**Figure 21.** Relative frequency distribution of the wake center $f$ in a reference frame centered at the rotor center.





Afterwards, the line starts to rotate with the wake and the relative frequency is concentrated at two diametrically opposite positions, similarly to what is observed by Li et al. (2022). Likewise, the line gradually smears out, until a diffuse pattern is

reached at $x^* = 7$, leading to more isotropic wake meandering. For SwHS, and as expected for a much lower sway amplitude, there is almost no difference with regards to the fixed-bottom case at $x^* = 1$. At $x^* = 3$, there appears to be some vertical meandering building-up, likely due to the meandering amplification combined with the wake rotation. From $x^* = 5$ onward, the pattern becomes circular and closer to that of the FB case.

The frequencies at which the wake center meanders are found through spectral analysis of both coordinates. For that, the

normalized amplitude spectrum defined below is computed:

$$A^*_{i\,\text{wc}}(St) = |\widehat{x^*_{i\,\text{wc}}(t)}|, \tag{13}$$

where $i$ denotes the considered direction, "wc" stands for wake center and the frequency is normalized ($St$). The results are depicted in Fig. 22 that follows the same row and column structure as Fig. 21.

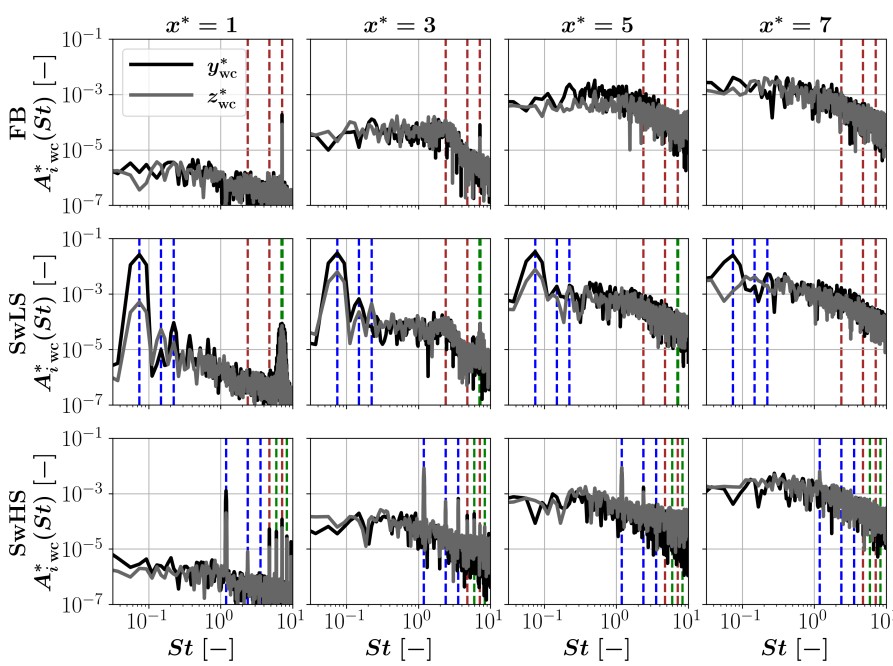

**Figure 22.** Amplitude spectra of the normalized wake center coordinates. The red lines are the first three multiples of the rotational frequency $St_\text{r}$. The blue lines are the first three multiples of the prescribed motion frequency $St_\text{p}$. The green lines are the sum and difference between $3St_\text{r}$ and $St_\text{p}$.

In general, the frequency content increases until $x^* = 5$ and starts plateauing at $x^* = 7$. The FB case initially presents with

very strong peaks at the 3P frequency but it loses dominance from $x^* = 3$ onward. For SwLS and SwHS, the prescribed motion frequency dominates all cases at all streamwise positions. The frequency interactions observed in Fig. 11 for the wake





velocity are also present in the meandering. A clear difference between the sway cases is that, for SwLS, the prescribed motion frequency is stronger and more latent in the wake as it progresses downstream.

Since the prescribed motion frequency dominates the meandering, the wake center normalized amplitude spectra at the prescribed motion frequency can be used to characterize the meandering intensity. Given that the meandering directions may rotate with the wake, it is more useful to characterize the intensity using a mean of both coordinates, such as the root mean square, as follows:

$$A^*_{\mathrm{wc}}(St_{\mathrm{p}}) = \sqrt{\frac{A^{*2}_{y_{\mathrm{wc}}}(St_{\mathrm{p}}) + A^{*2}_{z_{\mathrm{wc}}}(St_{\mathrm{p}})}{2}}. \tag{14}$$

If instead, one wishes to know how much the initial perturbation is amplified by the wake, the wake center amplitude spectra can instead be normalized by the normalized prescribed motion amplitude at the rotor center $A^*_{\mathrm{prc}}$, which yields the amplification factor $k$:

$$k_{\mathrm{wc}}(St_{\mathrm{p}}) = \frac{1}{A^*_{\mathrm{prc}}}\sqrt{\frac{A^{*2}_{y_{\mathrm{wc}}}(St_{\mathrm{p}}) + A^{*2}_{z_{\mathrm{wc}}}(St_{\mathrm{p}})}{2}}. \tag{15}$$

Figure 23 shows both $A^*_{\mathrm{wc}}(St_{\mathrm{p}})$ (a) and $k_{\mathrm{wc}}(St_{\mathrm{p}})$ (b), for all cases.

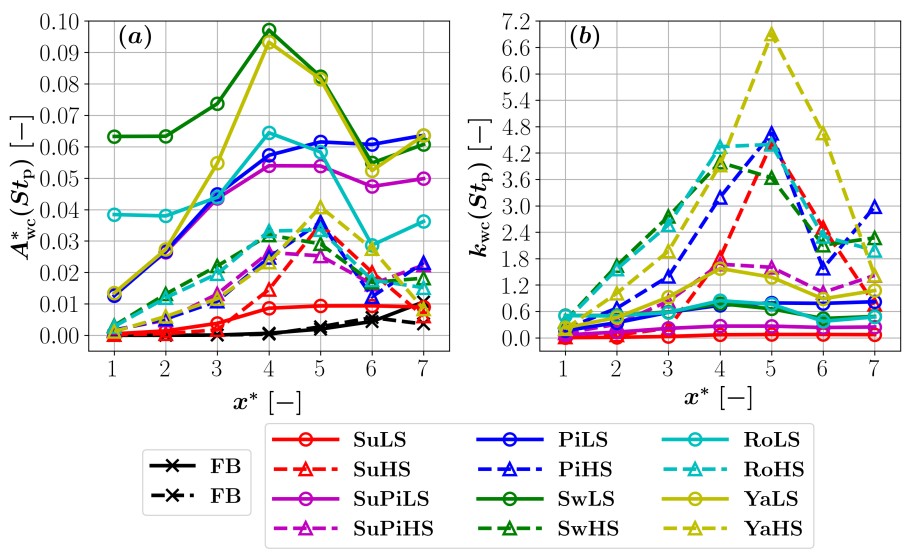

**Figure 23.** (a) Normalized average amplitude spectra; (b) amplification factors.

The FB case is excluded from the $k_{\mathrm{wc}}(St_{\mathrm{p}})$ plot since there is no prescribed perturbation. Regarding (a), the low-$St$/high-$A^*$ cases show the largest amplitudes across the domain, as expected. Curiously, the amplitudes dip at $x^* = 6$ and recover at $x^* = 7$, which suggests a transition to the natural meandering of the wake from the perturbation caused by the prescribed motion. This suggestion is supported by the fact the value for the FB case keeps increasing in that region. The surge case was an exception in this simulation group because the meandering is expected to be largely in the streamwise direction which cannot





be captured with the current method. Coupling surge to pitch tends to decrease the meandering amplitude beyond $x^* = 3$, when compared to pitch alone. For the high-$St$/low-$A^*$, the meandering intensity is smaller and peaks more downstream. The behavior from $x^* = 6$ is more heterogeneous with no clear trend. As for the amplification factors and similarly to Fig. 12, the high-$St$/low-$A^*$ cases are more amplified than the low-$St$/high-$A^*$ cases, once again with yaw dominating.

## 4 Conclusions

This paper addressed the wake dynamics of a laboratory-scale wind turbine under prescribed motions that covered five degrees-of-freedom and a coupled motion. The values of the prescribed amplitudes and frequencies were chosen to be realistic values for surge and pitch. The same values were used for the remaining analogous degrees-of-freedom so as to provide a fair comparison between cases and to investigated the potential effect of designing for non-realistic values of the remaining DOFs. The simulation setup replicated the POLIMI wind tunnel cross-section. The turbine was operated in the vicinity of rated conditions, under constant and uniform inflow and constant rotor speed in order to highlight the effects of the prescribed motion on the wake evolution.

The results highlight clear differences between prescribing motions at two different pairs of Strouhal number and normalized amplitude that were denoted by low-$St$/high-$A^*$ and high-$St$/low-$A^*$. The first group (i.e. low-$St$/high-$A^*$) exhibited a behavior that was relatively close to the fixed-bottom case in terms of time-averaged wake, turbulence intensity, wake recovery, shed vortices and shear layer evolution. By contrast, the second group (i.e. high-$St$/low-$A^*$) presented with more irregular wake boundaries (as defined by the time-averaged wake and turbulence intensity), a faster recovery, larger coherent structures on the shed vortices that were deformed by the prescribed motion and faster shear layer expansion near the rotor. This contrast was demonstrated to be caused by the prescribed motion itself whose upstream perturbations were amplified by the wake as it progressed downstream. The amplification happened in particular at the tip and root shear layers, where the prescribed perturbations were found to be correlated with the faster expansion of the tip shear layer, which likely contributed to a faster recovery. A proposed explanation for the wake amplification preference for the high-$St$/low-$A^*$ pair was given based on the fixed-bottom case wake spectra. Up to $x^* = 5$, these spectra showed a region of higher power that contained the frequency of high-$St$/low-$A^*$ pair and did not contain the frequency of low-$St$/high-$A^*$ pair. Hence, a resonance phenomenon could be the basis of the observed differences. The spectra also revealed frequency interactions between the prescribed motion frequency and the rotor frequency that had essentially decayed at $x^* = 5$. In the current setup, the prescribed motion directionality was found to play a role in the wake dynamics. More specifically, the impact of prescribed motions with a component perpendicular to the flow was found to be larger than that of motions exclusively in the flow direction.

The prescribed motions also caused wake meandering, with larger amplitudes being naturally observed for the low-$St$/high-$A^*$ group. The relative frequency distribution of the estimated wake center was deformed by the prescribed motion, rotated with the wake and approached a diffused circular shape at $x^* = 7$. The magnitude of the amplification of flow perturbations at the rotor center along the wake was quantified by amplification factors relating either velocity or displacement perturbations.





As expected, the maximum factors were associated with the high-$St$/low-$A^*$ and ranged between $1.2 - 3.6$ for velocity and $1.8 - 7.2$ for amplitude, with the highest values being found for yaw.

A short-coming of not simulating the nacelle in the current setup was the unrealistic wake energisation from its core that may have hindered the wake recovery in surge and stimulated it for the remaining DOFs. As for the simulation or wind tunnel
testing of the turbine, one should pay attention to wake interactions with the domain boundaries, that occurred between $x^* = 3$ and $x^* = 5$ in the conditions of this investigation. Possible next steps comprise: including the nacelle in the simulations and check its impact on the wake development; prescribing realistic six-DOF prescribed motion in order to see if these coherent structures arise in this situation; performing similar cases with different levels of turbulence to test its effect on the coherent structures produced by the floating motion and lastly, simulating full-scale rotors in field conditions and comparing the results
with the laboratory-scale rotors to draw insights on the $Re$ sensitivity.

**Appendix A: Prescribed motion validation**

The prescribed motion implementation was validated against the theoretical solution of the motion of one point in one of the blades. Deriving the solution involves describing the rotor center coordinates as a function of the prescribed motion and then summing the blade point motion relative to the rotor center due to rotation. Under prescribed translation, the variation in the
coordinates of all the turbine points due to the prescribed motion is exactly the same as the prescribed motion that affects that coordinate. Under prescribed rotation, the points on the turbine experience a variation in coordinates that is proportional to the distance to the rotation axis. Recalling the prescribed motion expression in Eq. (2), the sign was negative for translation and positive for rotation, in accordance to Bergua et al. (2023). The coordinates in time for an arbitrary point in the blade on the reference frame $e_x$-$e_y$-$e_z$ (see Fig. 1), with an initial azimuth of $\theta_0 = 0 \, deg$ are shown in Table A1, where $\theta = \omega_r t$
is the azimuth of a point on the blade; $\omega_r = 2\pi f_r$ is the angular frequency of the rotor; $t$ is the time; $R$ is the point's radial position relative to the rotor center; $L_e = \sqrt{L_1^2 + L_2^2}$ is the distance from the rotor center to the pitch axis (where "e" stands for effective) and $\alpha_e = tan^{-1}(L2/L1)$ is the effective angle between the rotor center and the vertical direction.

**Table A1.** Blade point coordinates under single-DOF prescribed motion.

| DOF | Surge | Sway | Heave | Roll | Pitch | Yaw |
|---|---|---|---|---|---|---|
| $x(t)$ | $p$ | $0$ | $0$ | $0$ | $L_2 + L_e\sin(p - \alpha_e) +$ $R\sin(p)\cos(\theta)$ | $R\sin(\theta)\sin(p) +$ $L_2(1 - \cos(p))$ |
| $y(t)$ | $-R\sin(\theta)$ | $p -$ $R\sin(\theta)$ | $-R\sin(\theta)$ | $-L_1\sin(p) -$ $R\sin(\theta + p)$ | $-R\sin(\theta)$ | $-R\sin(\theta)\cos(p) -$ $L_2\sin(p)$ |
| $z(t)$ | $z_{rc} +$ $R\cos(\theta)$ | $z_{rc} +$ $R\cos(\theta)$ | $p + z_{rc} +$ $R\cos(\theta)$ | $z_p + L_1\cos(p) +$ $R\cos(\theta + p)$ | $z_p + L_e\cos(p - \alpha_e) +$ $R\cos(p)\cos(\theta)$ | $z_{rc} + R\sin(\theta)$ |





The validation was performed by comparing the normalized displacement calculated with the above expressions with the actual value from the simulations, by means of an error parameter. The normalized displacement $\Delta x_i^*$ was defined as:

$$\Delta x_i^* = \frac{\Delta x_i}{L_\mathrm{b}} = \frac{x_i(t) - x_{i0}}{L_\mathrm{b}}, \tag{A1}$$

where $i$ is the direction, $L_\mathrm{b}$ is the blade length and $x_{i0}$ is the initial value of the coordinate $x_i$. The normalized displacement indicates how much the coordinate changed relative to the blade length. The relative error $\epsilon$ was defined as follows:

$$\epsilon = \frac{\Delta x_i|_\mathrm{sim} - \Delta x_i|_\mathrm{theo}}{A_\mathrm{pb}}, \tag{A2}$$

where $\Delta x_i|_\mathrm{sim}$ is the simulation displacement, $\Delta x_i|_\mathrm{sim}$ is the theoretical displacement and $A_\mathrm{pb}$ is equal to the blade point motion amplitude caused by the prescribed motion exclusively. This way, the motion prescription error was measured relative to the prescribed amplitude. The blade point motion amplitudes are summarized in Table A2. For surge, sway and heave, they are simply the prescribed motion amplitude. For roll and pitch, they represent the motion amplitude (in length units) of a blade point at radial distance $R$ from the rotor center and $\theta(t) = 0$ deg (assuming the rotor is not rotating). For yaw, the same is true but assuming $\theta(t) = 90$ deg. The comparison was done for the point at $82\%$ of the span with an initial azimuth of 0 deg.

**Table A2.** Blade point motion amplitude caused by the prescribed motion exclusively.

| Cases | Surge | Sway | Heave | Roll | Pitch | Yaw |
|---|---|---|---|---|---|---|
| $A_\mathrm{pb}$ | $A_\mathrm{p}$ | $A_\mathrm{p}$ | $A_\mathrm{p}$ | $A_\mathrm{p}(L_1 + R)$ | $A_\mathrm{p}\sqrt{(L_1+R)^2 + L_2^2}$ | $A_\mathrm{p}\sqrt{L_2^2 + R^2}$ |

Figure A1 compares the theoretical and simulation coordinate values for the PiLS, PiHS, SwHS and YaHS cases. The last three cases were included because they showed the largest errors. Both rows represent the coordinates affected by the prescribed motion (e.g. $x$ and $z$ for pitch). Each column represents one case. The time was normalized by the prescribed motion period $T_\mathrm{p} = 1/f_\mathrm{p}$. Looking at the theoretical and simulation curves, it can be seen that the prescribed motion closely follows the theoretical prediction, which is something that happens for all cases (including the ones not shown). Similarly to PiLS, all the low-$St$/high-$A^*$ cases show an error very close to 0%. The largest errors occur for the high-$St$/low-$A^*$ cases but they are nevertheless below 5% of the prescribed motion at the given point. This small mismatch can be explained by the fact that the prescribed motion is obtained through velocity prescription which leads to integration errors and some lag. We support this argument by noting that the errors are larger when the slope of the curves is higher, i.e, when the coordinate velocity is higher. In terms of length, these errors are below 0.5 mm which is marginal.



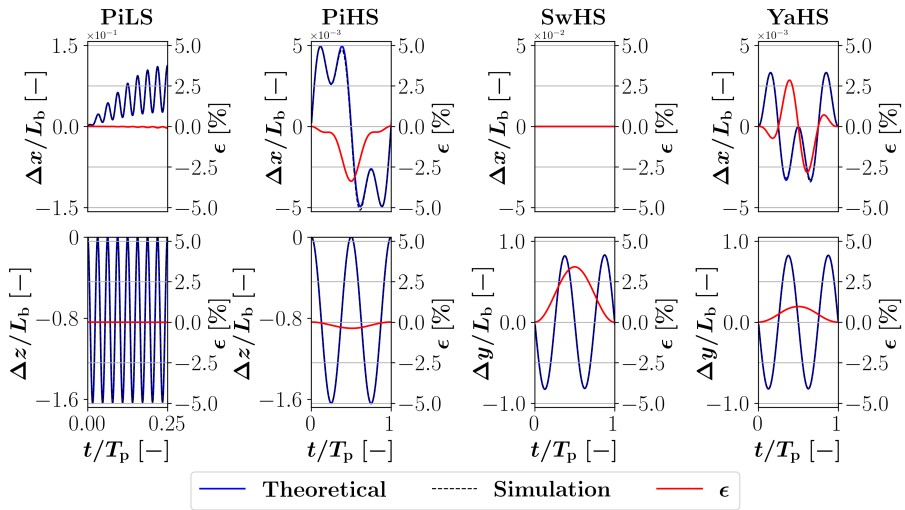

**Figure A1.** Difference between theoretical prediction of blade point motion and the simulation output.

## Appendix B: Flow field convergence

The flow field time convergence was assessed based on the convergence of the forward moving mean of the normalized streamwise velocity component $U_x/U_0$, defined by:

$$\frac{<U_x>}{U_0}(t) = \frac{<U_x>|_t^{t+\Delta t}}{U_0} \tag{B1}$$

and the forward moving standard deviation, defined by:

$$\frac{<U_x'^2>^{1/2}}{U_0}(t) = \frac{<U_x'^2>^{1/2}|_t^{t+\Delta t}}{U_0} \,, \tag{B2}$$

where $<\cdot>$ denotes the time average performed in the interval $[t, t+\Delta t]$, and $U_x' = U_x - <U_x>$. The simulations were performed for a period of time $T_{\text{sim}} = 64$ s such that the flow ran through the inlet until the end of the wake refinement region 8.6 times. The length of this portion was equal to $L_a^* = 12.5$. $T_{\text{sim}}$ was also equal to eight periods of the largest simulated frequency. The averaging time interval $\Delta t$ was equal to one prescribed motion period.

Figure B1 shows $U_x/U_0$ in black, $<U_x>/U_0(t)$ in blue and $<U_x'^2>^{1/2}/U_0(t)$ in green for the case SwLf. The first row and second rows stand for the azimuth position of $\theta = 0$ deg and radial positions $r^* = 0$ and $r^* = 1/2$ relative to the rotor center, respectively. Each column stands for a streamwise location. Similarly to SwLf in Fig. B1, all the simulated cases show a very reasonable convergence of the statistics at $t_{\text{conv}} = 4T_p = 32$ s (red dashed line). Hence, all the cases were deemed converged at this time and the remainder of the simulation, from 32 s to 64 s, was used for the post-processing.



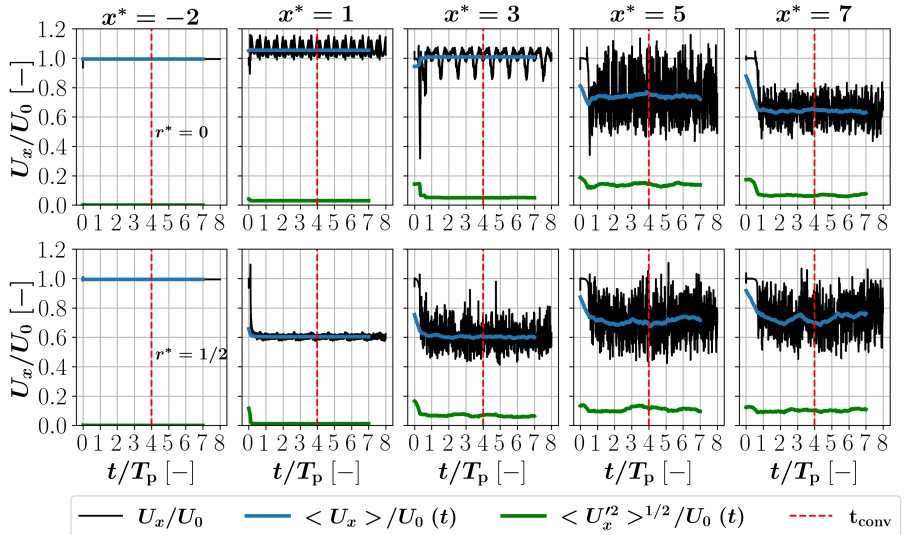

**Figure B1.** Streamwise flow velocity convergence for SwLf.

## Appendix C: Experimental comparison

Some cases were compared to the experimental data available in (Bergua et al., 2023; Cioni et al., 2023; Fontanella et al., 2021a, b). An extensive comparison between these data and the results of many participants was carried out with a focus on the loads in Bergua et al. (2023) and a focus on the wake in Cioni et al. (2023). Results for a setup similar to the one used in this paper are included in the latter. As for the rotor loads, the comparison is done in Fig. C1 for the thrust $T$ (top row) and torque $M$ (bottom row). Experimental data is available for the fixed-bottom (FB), surge cases (SuLS, SuHS) and pitch cases (PiLS, PiHS). The first column addresses the mean load value (denoted by ‾), the second column addresses the amplitude of oscillation (denoted by $'$) and the third column addresses the phase shift between the prescribed motion and the load $\phi_\mathrm{p} - \phi_\mathrm{load}$. The absolute value of the difference to the experimental value is denoted by $|\Delta|$. For the experimental cases, the mean value is taken over the full length of the time series. The amplitude and phase shift are obtained by computing the frequency content of a section of the time series using the FFT and extracting the values at the prescribed motion frequency. The section is the largest section possible such that the frequency step is a submultiple of the prescribed motion frequency. This fact together with the use of a flat-top window reduces the spectral leakage from sampling a finite time series, with the highest amplitude accuracy. The use of this window is also justified since a very well-defined and isolated peak is expected at the prescribed motion frequency. The same post-processing is performed for the simulations in the time series section after the transient period. Looking at the comparison, in terms of mean value, the simulations differ from the experimental results, at most, $8\%$ and $13\%$ for thrust and torque, respectively. On the other hand, the simulations show a mean value that is about the same in every case, which is what would be expected in the quasi-steady regime (Mancini et al., 2020; Fontanella et al., 2021b; Bergua et al., 2023), since all the





surge and pitch cases create relatively small variations of streamwise inflow velocity about the free-stream velocity (at most 5%).

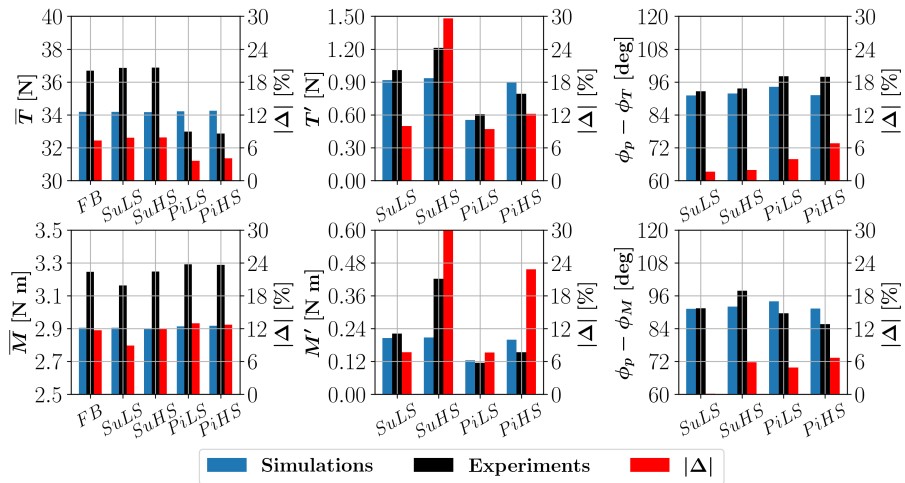

**Figure C1.** Thrust and torque comparison between simulations and UNAFLOW experiments.

Reasons for the mean value mismatch involve modeling the walls as slip walls, modeling the flow as laminar while it was slightly turbulent in the experimental setup and, arguably the most important factor, the high sensitivity that the actuator-line model has to the smearing length scale $\epsilon$ and the cell size $\Delta x$, as vastly reported in the literature (Mikkelsen, 2003; Troldborg et al., 2009; Martínez-Tossas et al., 2015, 2017; Amaral et al., 2024). It can be argued that the rotor loads can be adjusted by adjusting the previous parameters or by implementing model corrections (Meyer Forsting et al., 2019; Martínez-Tossas and
Meneveau, 2019) but the suitability of such strategies is still subject to investigation and implementation validation. Hence, and due to time constraints for further exploration of the topic, a uniform compromise value of $\epsilon/\Delta x = 2$ (Troldborg et al., 2009) was chosen for the used mesh resolution of $64/D$. Regarding the amplitude of the oscillations, the focus was made on SuLf, PiLf and PiHf, as SuHf was reported to have higher uncertainty in the measurements associated with the inertial loads (Bergua et al., 2023). The order of the differences is consistent with those observed for the mean value, since the amplitude of the load
variations is expected to scale with the steady-state value according to the quasi-steady theory (Fontanella et al., 2021b). There is however an uptick in the difference for PiHf that we attribute to the fact that the amplitude in the experimental setup (0.26 deg) was slightly lower than in the load case definition. As for the phase difference between the motion and the force, there is a good agreement between the simulation and the experimental results.

## Appendix D: Complete results



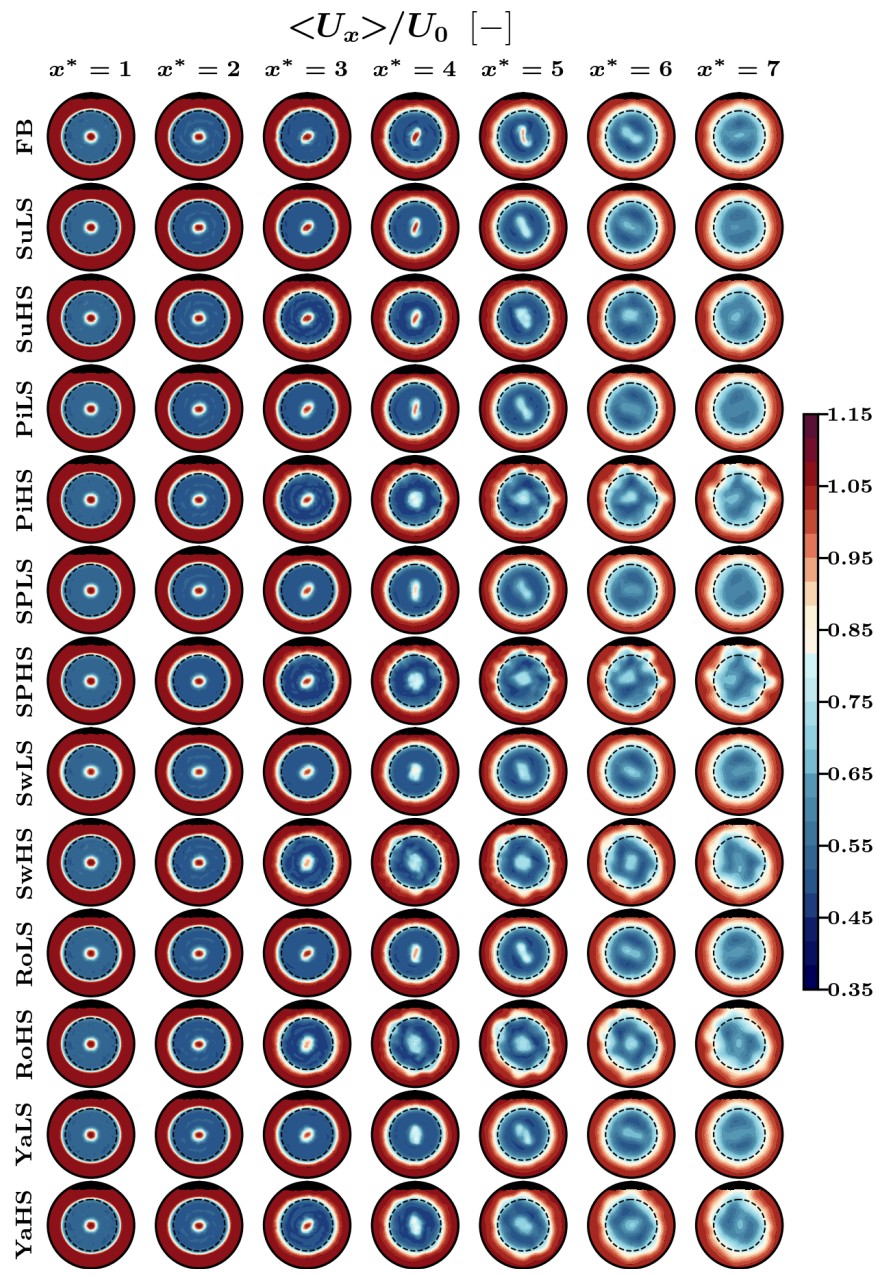

**Figure D1.** Normalized time-averaged streamwise velocity over the radial probes at several streamwise positions. See Table 3 for information about the cases. The black dashed circle indicates the rotor perimeter.



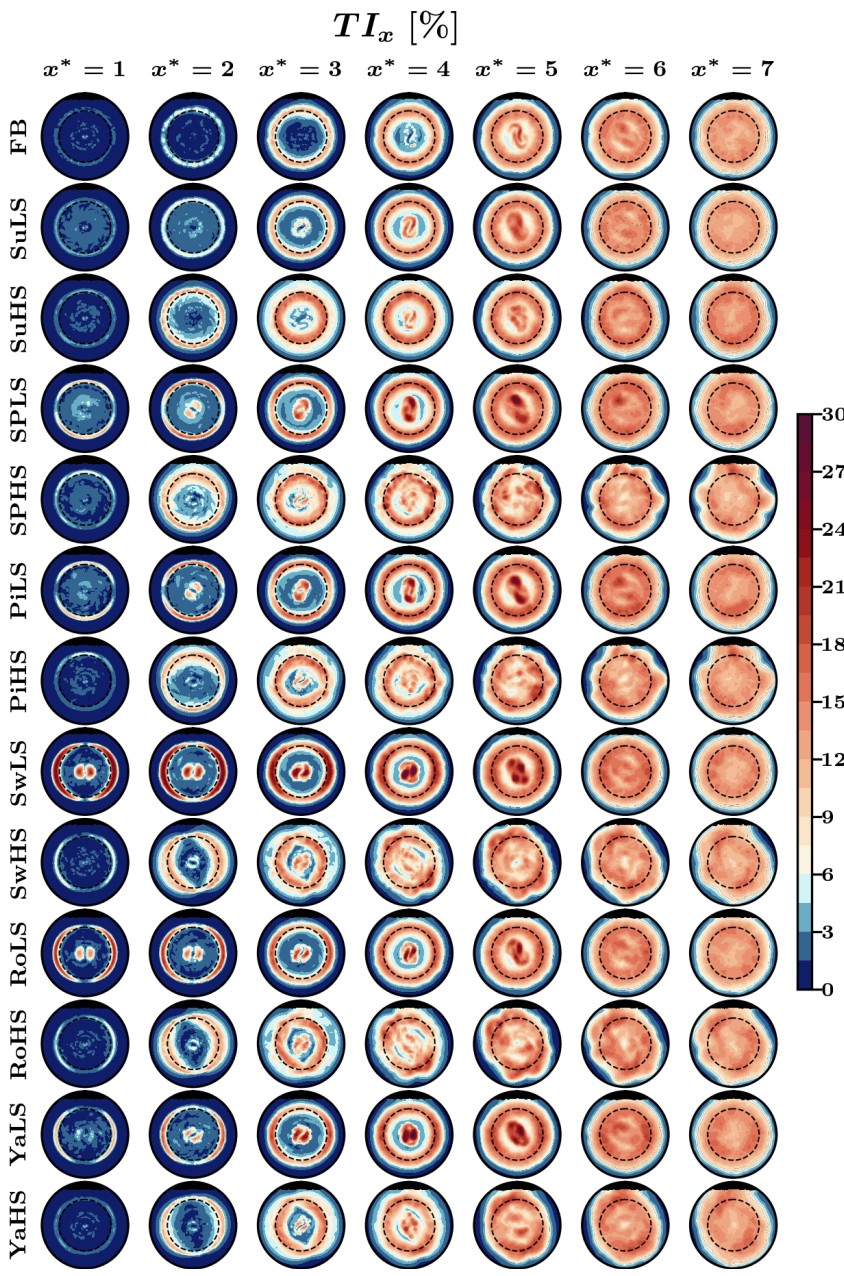

**Figure D2.** Streamwise turbulence intensity over the radial probes at several streamwise positions. See Table 3 for information about the cases. The black dashed circle indicates the rotor perimeter.

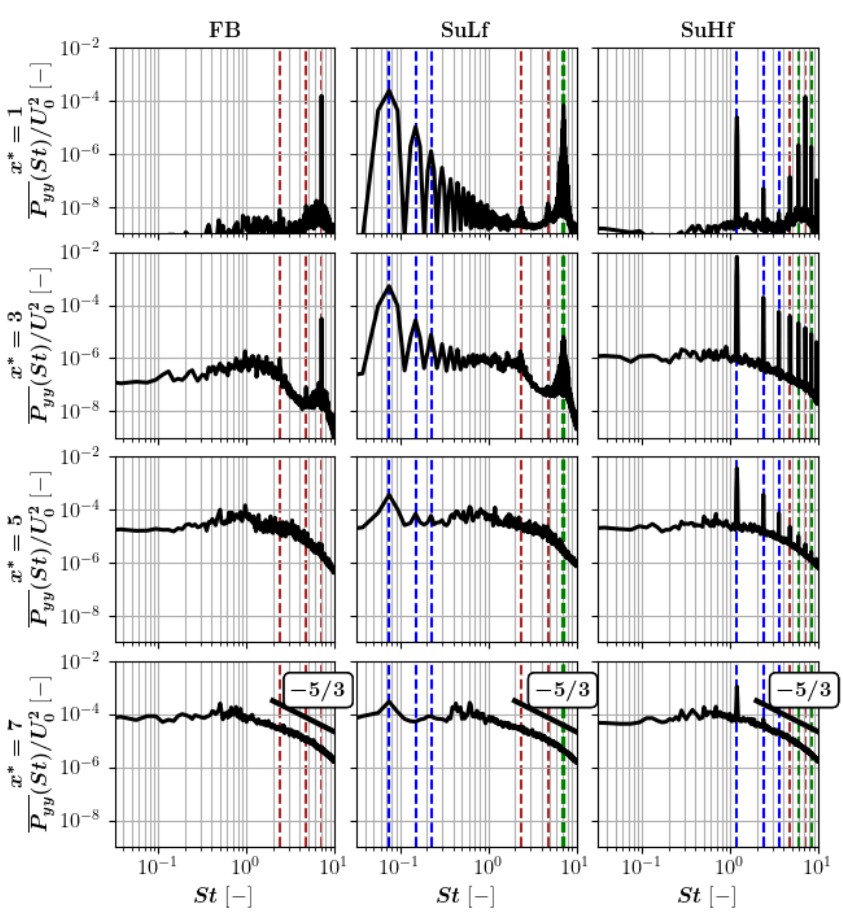

**Figure D3.** Spatially averaged wake spectra of the side-to-side velocity component. The blue lines are the first three multiples of the prescribed motion frequency $St_m$. The red lines are the first three multiples of rotational frequency $St_r$. The green lines are the sum and difference between $3St_r$ and $St_m$.

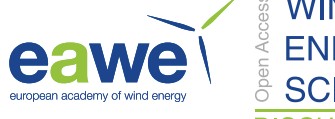

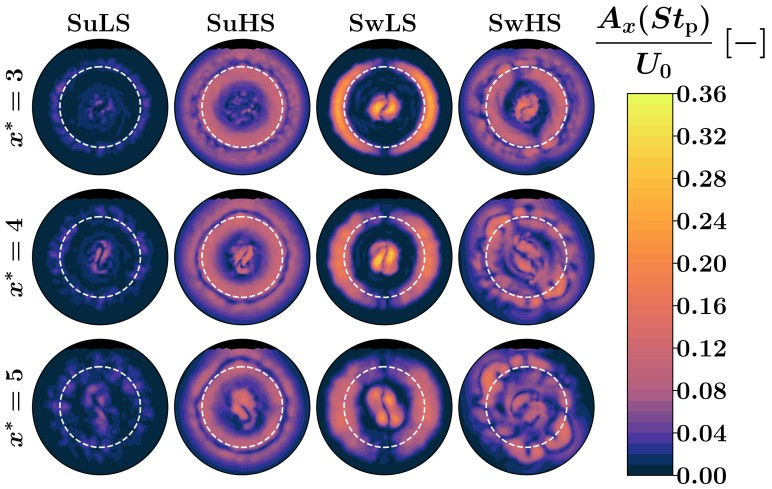

**Figure D4.** Normalized amplitude spectrum component of the streamwise velocity at the prescribed motion frequency over the radial probes for the surge and sway cases. The spectra for the remaining high-$St$/low-$A^*$ and low-$St$/high-$A^*$ cases are similar to those of the high-$St$/low-$A^*$ and low-$St$/high-$A^*$ sway cases, respectively.

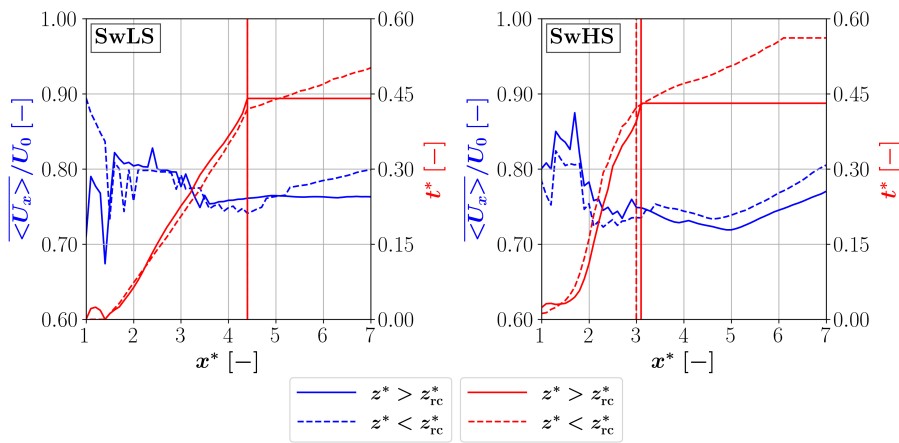

**Figure D5.** Normalized time-averaged streamwise flow velocity spatially averaged over the shear layer and tip shear layer thickness for the SwLS and SwHS cases in the plane $y^* = y^*_{\mathrm{rc}}$ (see Fig. 1).



*Data availability.* All simulation data is available upon request. Commercial licenses for YALES2 can be purchased from Laboratoire CORIA.

*Author contributions.* RA performed the simulations, post-processing and was responsible for writing the paper. FHM provided indispensable support in handling the simulation tool, provided text excerpts, references and some figures. All the authors provided valuable input and insights that were important to steer the work and write the paper. PD and AV were crucial is providing the necessary resources to produce
this work.

*Competing interests.* RA, DvT and AV declare that they have no conflict of interest. FHM, KL and PD declare that they were full-time employees of Siemens Gamesa Renewable Energy at the time this work was carried out.

*Acknowledgements.* This work results from the STEP4WIND project, a European Doctorate programme granted under the H2020 Marie-Curie Innovative Training Network (H2020-MSCA-ITN-2019, grant 860737.) We thank SURF (www.surf.nl) for the use of the National
Supercomputer Snellius. This work is part of W2ITASEC (Wind turbine Wake Interactions through Aero-Servo-Elastic Coupling) with computer resources provided by GENCI at TGCC (grant 2023-S142aspe00038 on the supercomputer Joliot Curie). We would like to thank Roger Bergua and Alessandro Fontanella for providing information regarding the experimental setup used to benchmark the simulations.





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
