# Peer review of "Investigating the dynamics of floating wind turbine wakes under laminar inflow using large eddy simulations"

_Wind Energy Science, 2025_

## Referee Comment (RC1)

**Review of Manuscript "Investigating the Dynamics of Floating Wind Turbine Wakes Under Laminar Flow Using Large Eddy Simulations**

by

R. Amaral, F. Houtin-Mongrolle, D. von Terzi, K. Laugesen, P. Deglaire, and A. Viré

**Overall Review**

This journal manuscript presents computational fluid dynamics results of wake flow from a simulated floating offshore turbine with prescribed motion over many degrees of freedom, such as pitch, sway, surge, etc. The simulation is of a wind-tunnel scale turbine in uniform inflow. The motion is completely prescribed and deterministic and not coupled with a hydrodynamics solver.

Two different forcing frequencies of the motion were simulated, and the results show that the higher simulated forcing frequency caused interesting and notable differences in how the wake becomes turbulent and then recovers. The authors point at this frequency possibly exciting a resonance effect.

The work is interesting and addresses the effect of floating platform motion on wakes. The conditions are very idealized, though, so I do wonder how extensible to realistic turbulent conditions the results are. The manuscript is very dense and long. It reads more like a technical report than a journal article. The manuscript would highly benefit from cutting down some of the very detailed explanation of results. I also have many specific comments below aimed at improving the paper. For these reason, I will select "reconsider for publication with major revision." See below for detailed comments.

**Specific Comments:**

- 1. Overall: The manuscript is long and dense. It would highly benefit from removing some non-essential material and making description of the results more succinct.
- 2. Page 2, Line 45: Here you talk about a low-frequency frequency content seen in wind turbine wakes associated with bluff body vortex shedding. Later on (and in the abstract), you talk about Strouhal number. You do not make an explicit connection between this wake meandering frequency and Strouhal number. Or do you mean something different by Stouhal number (see my comment 12 that I wrote after reading further).
- 3. Page 2, Line 47: Clarify what you mean by "counter-rotating shear layers." Do you mean that if looking within a plane cutting through the wake in the streamwise direction, the vorticity vector is opposite in the outer versus inner shear layers? Or are you referring to something having to do with wake rotation caused by rotor torque?
- 4. Page 3, Line 60: When talking about the work of De Cillis et al., you say that the tip and root vortices almost always transfer energy from the wake into the shear layer, slowing recovery. I do not really understand this statement. I think of the tip and root vortices as lying within the shear layer and that

they transfer energy from the wake to the outer flow and from the outer flow into the wake. I do not understand how energy would be transferred from the inner flow into the relatively thin shear layer itself.

- 5. Page 3, Line 63: What do you mean by "Kolmogorov-like flows?"
- 6. Page 3, Line 67: You say: "The research also established that the near-wake tip vortex sheet isolates the wake from the free flow, preventing momentum exchange and slowing down the wake recovery." How do you conclude that the tip vortex sheet somehow acts as a barrier between momentum inside and outside the wake? Is it not simply that even without tip vortices in the shear layer sheet (imagine an actuator disk in the flow), the small scale turbulence just doesn't efficiently transfer momentum relative to the Kelvin-Helmholtz-like rolls that take some distance to form at the wake edges and the subsequent strong turbulence that the rolls break down into that are very efficient in transferring momentum from inside the wake to outside and vice versa?
- 7. Page 3, Section 1.2: In my first pass through this section, I finished feeling confused. There are so many facts given from a lot of different sources that it becomes difficult for the reader to synthesize all of this into a coherent background. Also, some of the sentences are confusing, for example this one: "The mechanism identified as the enhanced recovery drivers were meandering structures for sway." I find this a confusing sentence because I don't understand what you mean by "meandering structures for sway." When I think of meandering structures, I think of the wake meandering horizontally vertically. When I think of "sway," I think of the turbine/floater moving side to side. I can see a connection between moving a turbine side to side which would give the wake an initial side-to-side meandering that might become amplified.
- 8. Page 3, Section 1.2: It would be helpful to split this section into paragraphs. Each time you talk about the research of another research group, start a new paragraph.
- 9. Page 3, Line 73: You say: "The perturbations in the domain were maximized for certain pairs of inflow perturbation amplitude and frequency." Please clarify what you mean. When I read this, it sounds like perturbations are maximized as a function of amplitude and frequency, but then it is unclear what is being maximized. You then go on to say "For those pairs, the dominant frequency, ..., was the inflow frequency." Again, this is unclear. What dominant frequency? What do you mean by "inflow frequency." My point is that without going and getting Mao and Sørensen's paper, I am having a hard time visualizing their numerical experiment.
- 10. Page 4, Line 118: In the sentence, "All these investigations demonstrate that upstream periodic perturbations within certain frequency and amplitude ranges modulate the wake by amplifying its meandering and anticipating the shear layer breakdown," I do not think the word "anticipating" is the right word choice. Maybe "accelerating" is the correct word.
- 11. Page 5, Line 144: How is the solver 4th-order spatially when it uses an unstructured mesh with finite volumes?
- 12. Page 8, Equation 3: With this definition of Strouhal number, it made me question my interpretation of Strouhal number in the earlier part of this paper. In the earlier part, you just refer to it as St, not  $St_p$ , and I was thinking you meant a Strouhal number of wake meandering (you refer to bluff-body wake behavior). But now I question myself. Please make all of this clear and consistent to the reader.
- 13. Page 7-8: Your variables have various subscripts, such as "prc" or "rp." I assume "p" means "prescribed" but it is not explicitly said for all these letters. The reader should not have to guess.
- 14. Page 8, Line 186: I don't understand what the length and velocity scaling factors,  $\lambda_L$  and  $\lambda_U$ , are used for. Please clarify.
- 15. Figure 6: Why is the top of each contour plot cut off (it appears as black)?

- 16. Page 11, Line 235: You say, "...excludes the central jet that does not occur in reality." I think it is important to remember that the "central jet" does not form just because the simulation lacks a hub/nacelle, but more because the inner sections of the blade do not apply much thrust force on the flow, so these inner sections do not act very much as a momentum sink, especially where the blade section is circular. So, it is not true to say that the inner jet does not occur in reality.
- 17. Page 11, Line 243: I do not understand this quantity "excess wake recovery speed." Please elaborate more about what it is and why it is relevant. Is it just a normalized measure of the difference in velocity gradient between each case and the fixed bottom case, and you are saying that the steeper the gradient, the faster the wake recovery? What if it is a case in which the wake has recovered much faster than the FB case so at some point downstream its gradient then becomes less than in the FB case and the excess wake recovery speed becomes negative?
- 18. Figure 7 (c): What happens to excess wake recovery speed at x = 4 where all cases are shifted downward?
- 19. Section 3.2: I think it is important to distinguish between "turbulence intensity," which is just a statistic that is caused by true turbulence versus wake motion from prescribed rotor motion. Very near the rotor, before the wake has broken down into a turbulent state, the contour plots still show turbulence intensity which follows the prescribed motion (for example, you see it on the sides of the rotor disk for sway motion or on the top/bottom for pitch motion, but this is not really turbulence.
- 20. Page 16, Line 322: I do not understand what you mean where you say "|·| denotes the complex number modulus and the frequency f was nondimensionalized into St." Please clarify. I think you are saying two distinct things here: (1) The vertical bars mean that you take the FFT and then take the complex modulus of the real plus imaginary number you get at each frequency. (2) The frequency you plot spectrum against is replaced by Strouhal number because Strouhal number contains frequency. Is that correct? If so, please say it more explicitly.
- 21. Page 16, Line 330: In "These oscillations moderate relative," you are missing an "are."
- 22. Page 21, Line 415: You say, "The shear layer is a key aspect of the wake recovery because it isolates the wake from the free flow, hindering momentum entrainment and slowing the recovery down." I disagree that the shear layer acts as this isolating layer. There is nothing about it that isolates the wake flow and hinders momentum entrainment. Without a shear layer and its instability, which must be triggered with perturbations that grow with time/distance, the mixing would not occur. I think it is more accurate to say that "the shear layer is a key aspect of wake recovery because it feeds the instabilities that break down into turbulence that cause the momentum transfer that drives wake recovery."
- 23. Page 22: Figure 16: It is not totally clear what this figure shows. I am assuming you are tracing out the *y*-location of the inside and outside edge of the shear layer. Please say that very explicitly. I also do not think it right to say that the shear layer "growth rate sharply decreases at around  $x^* = 5$ ." As you say next, this coincides with the merging of the inner and outer shear layer, so the shear layer can no longer grow, hence the sharp decrease. I would just stop the inner line at the point of merger, and indicate that this is the point of merger.
- 24. Page 23, Line 454: I do not think it correct to say that the wake center jet is because you don't model the nacelle. Even with the nacelle, you'll still have a jet because of the aerodynamically unloaded inner sections of the blade.
- 25. Page 23, Line 456: You say, "When the merger happens, the shear layer is no longer capable of expanding inwards and only expands outwards, leading to the recovery." I do not understand how the shear layer only being able to expand outward itself causes recovery? Looking at the instantaneous flow field, it seems that shear layer merging happens because the shear layer instabilities have broken into full turbulence at that point, and only then is there sufficient mixing to get wake recovery underway.

- 26. Page 24, Line 498: You say, "The amplitude in the *y*-direction is much larger than that in the *z*-direction but no considerations can be drawn from this fact since the wake is rotating." What does wake rotation have to do with this? Also, can't it be explained by the lower and upper walls constrain the *z*-motion, but the wake is free to meander in the *y*-direction?
- 27. Appendix A: You should state up front that you derive position by integrating velocity. Because that fact comes at the very end, I found myself stopping to see if I had missed something because I did not understand where the error comes from. If you state that position is derived up front, then you give the reader context as to why you do this error analysis.
- 28. Appendix B: It is not clear to me how you did averaging. Are you saying that you time averaged over the second half of the simulation? Your blue and green lines indicating mean and variance appear more as moving averages than the convergence of the average over the whole time period. You indicated a forward moving average, but you also say that you use the second half of the simulation for averaging. By "converged" do you mean the initial transients are gone or the averaging length is long enough.
- 29. Appendix C, Line 633: Where you say, "An extensive comparison between these data and the results of many participants," who are the many participants?
- 30. Appendix C, Line 634: Where you say, "Results for a setup similar to the one used in this paper are included in the latter," what is "latter" referring to. It would be better to just explicitly say it instead of using "latter."
- 31. Appendix C, Line 635: Where you say, "Experimental data is available...," it should be "Experimental data **are** available..." The word data is plural, so just do a search for the word "data" in the manuscript and make sure the verb reflects this.
- 32. References, Line 745: Hojstrup should be Højstrup.

---

## Referee Comment (RC2)

**Review:** Investigating the dynamics of floating wind turbine wakes under laminar inflow using large eddy simulations**

**General Comments:**

This paper uses LES of a lab-scale DTU10MW turbine to study the wake behaviour of a single floating wind turbine under a number of different prescribed motion cases, using two Strouhal number and amplitude combinations per case. It is shown that high Strouhal number motions lead to a faster wake recovery, and that prescribed motions with the flow-perpendicular components have a larger impact than those only in the streamwise direction. This topic is an interesting and novel field of research that is relevant to the Wind Energy Science journal.

However, the paper has some serious flaws. The most concerning is the extremely small domain size (specifically in the frontal area) which results in a very high blockage ($\approx 8.5\%$) which will substantially influence the results, particularly when a prescribed motion is being studied. The results provided in the paper show clear evidence of this, but it is very rarely mentioned or explained. For example, there is a cut-off at the top of all cross-flow wake plots because the domain is so small (see Figures 6, 9 and 13), and the interaction of the wake with the top boundary of the domain appears to impact the results in many other plots (see Figures 7b, 19 and D5). Lines 465- 475 (near the end of the results) is the first place where the impact seems to be noted, with comments such as 'the shear layer near the top wall interacts with it relatively upstream', and 'the shear layer growth stops upon the interaction with the top boundary'. The justification 'it is still relevant to observe it because real floating turbine wakes are effectively bounded by the water on one side and may, therefore, present this particularity' suggests a fundamental misunderstanding of what is occurring in results (which is a purely setup-induced issue) versus what a real floating turbine wake would experience. The reader only learns in the first paragraph of the conclusions that the computational setup is supposed to replicate the POLIMI wind tunnel. However, only in Appendix C is any comparison with experiments shown (only thrust and torque), and the paper overall is presented not as a validation against experiments but as a novel piece of research on floating wind turbine wakes. Therefore, the influence of the domain size must tested by rerunning a subset of the simulations in a larger domain with significantly reduced blockage ($< 3\%$), with the option to rerun all simulations if necessary, or the paper should be rewritten to be a comparison with experimental data and therefore foreground the comparison with the POLIMI wind tunnel.

In addition, the paper is extremely long and should be made more concise, with some figures and descriptions removed. Many of the key simulation setup details are missing, unclear or only found in the Appendices, and hence the results would be very difficult to reproduce. Some of the key choices in setup (Strouhal numbers, domain size, laminar flow, lab-scale turbine) are not justified or are only mentioned in the conclusions. The paper would also benefit from a discussion section before the conclusions in which the main findings are clearly stated and compared to other work, and the shortcomings of the study explored.

Therefore, in light of the need to run a substantial number of additional simulations and conduct significant restructuring and rewriting of the paper, I cannot support publication.

See specific comments below.

**Introduction:**

General comments: Overall I think that the introduction lacks sufficient motivation for the study, and the research gap that the paper seeks to fill is not sufficiently clarified. I think it would be preferable to remove some of the more lengthy descriptions of previous work and focus more on interpretation and clarification of what novelty this work provides.

Suggested additional references: Gupta and Wan 2019, 'Low-order modelling of wake meandering behind turbines' (regarding meandering and amplification of inflow perturbations); Xu, et al., 2023, 'Numerical Investigation of Aerodynamic Responses and Wake Characteristics of a Floating Offshore Wind Turbine under Atmospheric Boundary Layer Inflows' (regarding influence of ambient TI on FOWT wakes); Hodgson et al. 2023 'Effects of turbulent inflow time scales on wind turbine wake behavior and recovery' (regarding interaction between inflow scales and wake recovery/spectra).

**Methodology and Setup:**

Line 137: Personally not sure that stating the Navier-Stokes equations adds much.

Line 140: Reference needed for the actuator line model - Sørensen and Shen, 2002 'Numerical Modelling of Wind Turbine Wakes'.

Line 143: You say that the AL is included as a velocity source term, is that what is usually done? I think it is normally applied as a force term, not a velocity, so perhaps this requires clarification.

Line 146: Why use an unstructured grid when you are using the AL which is designed to work in structured grids? Will this have an impact on your results?

Line 150: It is unclear why you are modelling a scale model of the DTU10MW turbine, rather than just the 10MW turbine itself, seeing as you are using LES. From the results in the Appendices I assume that it is to do with comparison with experimental data, but the experimental data is not actually used in this paper, and there is no explicit justification of this choice anywhere.

Line 176: There is a lack of justification of the use of laminar flow - the Introduction section implies that this can significantly affect the results (e.g. Messmer et al., 2024), and one of the reasons stated in the Appendix for the discrepancies with the experimental data is that there was some TI in the experiment.

Line 180: I think it is fine to include a fixed bottom case, but in general do you think it is an appropriate baseline to compare to? In the prescribed motion cases, energy is being added to the system through the turbine motion, whereas it is not for the fixed bottom case.

Line 183: Further justification of your choice of Strouhal numbers is required, also in relation to what has been studied in the literature. The two values chosen are very high and very low compared to previous work, so it should be made clear why and how this choice was made. Also, only considering two points makes it very difficult to see trends in results.

Line 198: I have serious concerns about the domain size used in the simulations. It seems clear the blockage will be an issue as your blockage is 8.5% which is significantly larger than the recommended < 3%. In addition, I assume that this is why there are black cut-off regions at the top of all wake plots and why some plots of vertical shear layer expansion are not included as there is interaction with the top of the domain.

Line 198: There is no justification stated for this domain size choice, which seems to be the comparison with the POLIMI wind tunnel, but this is only stated in the first paragraph of the conclusions.

Line 203: Report the mean grid resolution in the refined region, not the maximum. Also report the size of the refined region in the text.

Line 206: After examining the results presented in the Appendix, I am not sure about the claim that the setup has been 'validated extensively'. There are discrepancies in thrust and torque in Figure C1 and the only explanation given is the slight turbulence and the smearing parameter of the AL. I understand that matching experimental data is challenging, but I think that presenting these plots as a comprehensive validation is misleading.

Line 210: Where are the numerical details of your simulation? Total time, time over which data is analysed, time step, details of the AL method including smearing parameter etc? I see that many of these are written in the Appendix but they should be written in the methodology section.

**Wake:**

General comment: This section should be named Results.

Line 218: State the averaging time and total time in the methodology, the reader should not have to refer to the Appendix to obtain key details of the setup. Also clarify the method used, e.g. in Appendix B you refer to eight periods of the 'largest simulated frequency', is this the lowest or highest frequency?

Line 231: In my opinion the paper becomes a lot less readable when a huge amount of results are presented in the Appendix and then a small subset is reproduced in the main text, as it becomes very confusing - is the reader supposed to look at all results to see the conclusions you draw or just the subset given? For example, in Figure 6 I am not sure that the extra wake excitement for high St flows is visible except for the SwHS case. I would prefer removing the full results from the paper and presenting a clear and concise subset instead.

Line 250: Could the behaviour after $x* = 4$ (e.g. a drop/ flattening of the relative wake recovery) be due to interaction with the domain boundary as the wake expands downstream?

Line 257: Please justify why your Strouhal numbers are so different to previous work rather than just stating it.

Line 294: In Figure 9, does the increased turbulence intensity around the wake edge at $x* = 2$ not just reflect the movement of the turbine, and hence provide a quite biased view of implications for the wake behaviour?

Line 304: In Figure 10, clarify whether the percentages in Figures b) and c) are relative or absolute, as the unit on all three plots is percent.

Line 306: 'The link makes sense because there is more momentum available for entrainment...' Not entirely sure what you mean by this sentence, needs clarifying or removing.

Line 369: In Figure 12, can you clarify why there are two lines (both full and dashed) for the FB case?

Line 423: Can you explain more clearly how you created the plots in this section, when the position of the turbine (and then consequently the position of the near wake) is moving? And how does the turbine motion affect these plots?

Line 429: Overall I find this section, in particular Figures 16-18, to be quite confusing and difficult to interpret, as it seems to be based on an arbitrary threshold for defining the shear layer thickness and is used somewhat inappropriately as a proxy for energy entrainment or momentum transfer into the wake. I would suggest instead looking at energy transfer through reynolds stresses, mean kinetic energy transport or TKE advection (all of which can also help in defining the near wake

length) - see Wu and Porte-Agel 2012 'Atmospheric turbulence effects on wind-turbine wakes: An LES study'.

Line 454: The few sentences starting with 'The explanation for the merger-recovery coincidence is also intuitive'. You seem to be implying that the wake only starts recovery after the shear and hub vortices are merged, and specifically starts recovering because it 'can only expand outwards'? I don't think this is a correct description of near wake behaviour and at best is extremely speculative, so this should be removed.

Lines 465 - 475: I find these section extremely worrying regarding your simulation setup. Statements such as 'the turbine is closer to the top wall than to the bottom wall', 'the shear layer near the top wall interacts with it relatively upstream' and 'the shear layer growth stops upon the interaction with the top boundary' seem to provide evidence for my concerns in the simulation setup that your computational domain is far too small and that this significantly impacts your results. The justification that 'real floating turbine wakes are effectively bounded by water on one side and may, therefore, present this particularity' I find to be wholly unconvincing. The real world implications of having a ground plane below the turbine (where the velocity is zero, as there is roughness in reality) are completely different to artificially introducing a slip boundary in the high-velocity flow directly above the turbine.

Line 488: Figure 19 seems to clearly highlight that the domain is too small to properly examine the wake flow?

Line 498: 'The amplitude in the y direction is much larger than that in the z direction but no considerations can be drawn from this fact since the wake is rotating'. Isn't this just because the prescribed motion is lateral?

Line 534: I don't think that having amplitude plots for all three of wake velocity, shear layer velocity and wake centre velocity are necessary, particularly when no effort is made to compare them.

**Conclusions:**

General comment: This paper needs a discussion section before the conclusions. In a discussion section you should give a summary of your main findings (and therefore can shorten the conclusions to be more concise), discuss the implications of those findings, and then discuss the shortcomings of your work and give justifications. For example - the choices of Strouhal number, the choice to have laminar and uniform inflow, to model a lab-scale turbine, etc. These should not just be left to a few sentences at the end of the conclusions but need to be explored in detail both in the simulation setup and in a discussion section.

**Appendices:**

General comment: As mentioned in previous comments, key details of the simulation setup should not only be stated in the appendices, but must be included in the main body of the text.